# MULTIDIMENSIONAL TRAJECTORY OPTIMIZATION FOR FLOW AND DIFFUSION

## ABSTRACT

In flow and diffusion-based generative modeling, conventional methods rely on unidimensional coefficients for the trajectory of differential equations. In this work, we first introduce a multidimensional coefficient that generalizes the conventional unidimensional coefficient into multiple dimensions. We also propose a new problem called multidimensional trajectory optimization, which suggests a novel trajectory optimality determined by the final transportation quality rather than predefined properties like straightness. Our approach pre-trains flow and diffusion models with various coefficients sampled from a hypothesis space and subsequently optimizes inference trajectories through adversarial training of a generator comprising the flow or diffusion model and the parameterized coefficient. To empirically validate our method, we conduct experiments on various generative models, including EDM and Stochastic Interpolant, across multiple datasets such as 2D synthetic datasets, CIFAR-10, FFHQ, and AFHQv2. Remarkably, inference using our optimized multidimensional trajectory achieves significant performance improvements with low NFE (e.g., 5), achieving state-of-the-art results in CIFAR-10 conditional generation. The introduction of multidimensional trajectory optimization enhances model efficiency and opens new avenues for exploration in flow and diffusion-based generative modeling.

## 1 INTRODUCTION

Flow and diffusion-based generative modeling (Song et al., 2021; Karras et al., 2022; Lipman et al., 2023) demonstrates remarkable performance across various tasks and has become a standard approach for generation tasks. We introduce the novel concept of the *Adaptive Multidimensional Coefficient* and propose an optimization problem termed *Multidimensional Trajectory Optimization* (MTO) in this field. As described by Albergo et al. (2023), the trajectory with $x_0 \sim \rho_0$ and $x_1 \sim \rho_1$ in flow and diffusion for $t \in [0, T]$ can be written as $x(t) = \alpha_0(t)x_0 + \alpha_1(t)x_1, \quad x_0, x_1 \in \mathbb{R}^d$, where conventionally, the

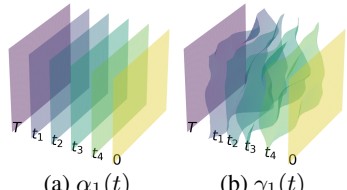

(a) $\alpha_1(t)$      (b) $\gamma_1(t)$

Figure 1: Comparison between unidimensional ($\alpha_1(t)$) and multidimesional ($\gamma_1(t)$) coefficient.

coefficients $\alpha_0(t), \alpha_1(t) \in \mathbb{R}$ are unidimensional. We extend this by introducing a multidimensional coefficient $\gamma_0(t), \gamma_1(t) \in \mathbb{R}^d$, allowing different time scheduling across all data dimensions.

By leveraging the increased flexibility provided by the multidimensional coefficient, our multidimensional trajectory optimization addresses the key question: *"Given a differential equation solver with fixed configurations, which multidimensional trajectory yields optimal performance in terms of final transportation quality for a given starting point of the differential equation?"* This question highlights a trade-off inherent in diffusion models, where simulation-free objectives—though beneficial in reducing training costs—limit adaptability in trajectory optimization concerning output quality, a flexibility retained in simulation-dynamics (Chen et al., 2018). To enable trajectory optimization while maintaining simulation-free objectives for training cost, prior approaches have relied on pre-defined trajectory properties, such as straightness (Liu et al., 2023; Tong et al., 2024), to minimize numerical error. However, such pre-defined properties for the trajectories diverge from true optimality in transportation, as they do not account for the sole measure of optimality which can be calculated by simulation-dynamics in our perspective: the final quality of transportation.

We reintroduce trajectory adaptability by employing simulation dynamics combined with adversarial training (Goodfellow et al., 2014), defining trajectory optimality based solely on the final generative output under fixed solver configurations. Specifically, first, we pre-train a diffusion model $H_\theta$ with randomly sampled multidimensional coefficients $\gamma$ from a well-designed hypothesis space. This pre-training enables the flow or diffusion model to handle various coefficients, preparing it for the trajectory optimization stage. Next, we introduce the parameterized adaptive multidimensional coefficient $\gamma_\phi$ to compose a flow or diffusion-based generator $G_{\theta,\phi}$, which produces $x_{\text{est},\theta,\phi}$ through simulation dynamics. A discriminator $D_\psi$ evaluates the generated samples $x_{\text{est},\theta,\phi}$ to optimize both $\theta$ and $\phi$, a process we term multidimensional trajectory optimization.

To effectively leverage the advantages of simulation-based objectives which lie on adaptability and flexibility of trajectories while mitigating their inefficiencies in training, we use simulation-based objectives only for $\phi$ after pre-training $\theta$ with a simulation-free objective, making trajectory optimization feasible in terms of efficiency and scalability. Our experiments demonstrate that trajectories optimized through this approach significantly improve the performance of flow and diffusion models. In summary, our **main contributions** are as follows:

1. By introducing the concept of an adaptive multidimensional coefficient in flow and diffusion, we lay the groundwork for complete trajectory flexibility.

2. We address a novel problem—multidimensional trajectory optimization—leveraging the increased flexibility provided by the adaptive multidimensional coefficient. This introduces an optimality concept based solely on the end-to-end final transportation quality rather than on pre-defined properties of the trajectory.

3. We propose a solution to the multidimensional trajectory optimization problem using adversarial training to discover adaptive multidimensional trajectories for efficient inference.

By introducing adaptive multidimensional trajectories, our work alleviates the constraints on trajectories in flow and diffusion models and opens new avenues for future research and applications.

## 2 RELATED WORKS

**Trajectory Optimizations in Flow and Diffusion**   Various trajectory optimization approaches pre-define optimality without relying on final transportation quality. Approaches such as Liu et al. (2023); Tong et al. (2024) define straightness as the optimality criterion and optimize trajectories by maintaining the consistency of $(x_0, x_1)$ for training flow and diffusion models, aligning with an optimal transport perspective. Another example is Singhal et al. (2023), which defines optimality through a fixed sequence of diffusion steps intended to reduce inference complexity rather than focusing on the quality of final samples. Additionally, Bartosh et al. (2024) introduces neural flow models that implicitly set trajectory optimality within the diffusion process, aiming to generate high-quality samples without explicit trajectory adjustment post-training. There are also approaches that refine trajectories after training, such as Albergo et al. (2024), where optimality is defined by minimizing the trajectory length in the Wasserstein-2 metric, focusing on a shortest-distance criterion. Despite these diverse perspectives on trajectory optimality, there are two significant differences between these methods and ours. First, we calculate the optimality of the trajectory solely based on the final transportation quality, which is a crucial factor in generative modeling. Second, none of these methods achieve full flexibility of the trajectory on two fronts: multidimensionality and adaptability with respect to different inference trajectories.

**Diffusion Distillation for Few-Step Generation**   There are two main approaches to diffusion distillation: non-adversarial and adversarial. Non-adversarial methods, like Yin et al. (2024); Song & Dhariwal (2023); Geng et al. (2024); Berthelot et al. (2023), focus on 1-step distillation techniques without adversarial objectives. These approaches aim to simplify training by leveraging distributional losses and equilibrium models, effectively distilling the diffusion process without involving a GAN discriminator. Conversely, adversarial approaches to diffusion distillation, such as Zheng & Yang (2024); Xu et al. (2024); Wang et al. (2023), employ a GAN-like discriminator to enhance sample quality by learning distribution consistency in an adversarial setting. Additionally, Luo et al. (2024) propose Diff-Instruct, which transfers knowledge from pre-trained diffusion models through a GAN-based framework, closely resembling GAN approaches in training dynamics. Also, Kim et al. (2024) developed Consistency Trajectory Models (CTM), which generalize Song et al. (2023) for efficient sampling with the assistance of a discriminator. While our method shares similarities in achieving few-step generation, there is a key difference: distillation in these works is not aimed

at trajectory optimization. Our method optimizes both $\theta$ and $\phi$, representing the diffusion model parameters and adaptive multidimensional coefficient parameters, with respect to different inference trajectories, thereby enabling adaptive multidimensional trajectory optimization. In contrast, distillation typically targets a fixed teacher network, limiting flexibility in optimizing the trajectory.

## 3 PRELIMINARY

We consider the task of transporting in two distributions $x_0 \sim \rho_0$ and $x_1 \sim \rho_1$, where $x_0, x_1 \in \mathbb{R}^d$. Following Albergo et al. (2023), for $t \in [0, T]$, **the trajectory** $x(t)$ is:

$$x(t) = \alpha_0(t)x_0 + \alpha_1(t)x_1, \quad v(t, x(t)) = \dot{\alpha}_0(t)x_0 + \dot{\alpha}_1(t)x_1, \quad \alpha_0(t), \alpha_1(t) \in \mathbb{R}, \quad (1)$$

where $\alpha(t) = [\alpha_0(t), \alpha_1(t)]$ represents the unidimensional-valued coefficients and $\dot{\alpha}$ denotes the derivative of $\alpha$ with respect to $t$. Diffusion models the vector field $v(t)$ as follows:

$$[x_0, x_1] \approx [\hat{x}_{0,\theta}, \hat{x}_{1,\theta}] = \mathrm{NN}_\theta(t, x(t)), \quad v(t, x(t)) \approx \hat{v}_\theta(t, x(t)) = \dot{\alpha}_0(t)\hat{x}_{0,\theta} + \dot{\alpha}_1(t)\hat{x}_{1,\theta}, \quad (2)$$

where $\mathrm{NN}$ denotes a neural network. For example, Song et al. (2021) predict the score value $\nabla \log p(x(t); t) = -\hat{x}_{1,\theta}/\alpha_1(t)$ to obtain the vector field. There are also flow-based methods, such as Lipman et al. (2023), that do not explicitly target $\hat{x}_{0,\theta}$ or $\hat{x}_{1,\theta}$ but instead directly model the vector field $\hat{v}_\theta(t, x(t)) = \mathrm{NN}_\theta(t, x(t))$. All these methods achieve generative modeling by numerically solving an ODE or SDE using the predicted vector field $v_\theta(t, x(t))$. In this section, we introduce two specific methods utilized in our experiments.

**Elucidating Diffusion Model (EDM)**   The Elucidating Diffusion Model (Karras et al., 2022) refines and stabilizes diffusion model training, using $x_0 \sim \rho_0$ as data and $x_1 \sim \rho_1 = \mathcal{N}(0, I)$. The coefficient is defined as:

$$\alpha(t) = [\alpha_0(t), \alpha_1(t)] = [1, t], \quad T = 80. \quad (3)$$

EDM minimizes $\|H_\theta(t, x(t)) - x_0\|_2^2$ for $H_\theta$ where $\hat{x}_{0,\theta} = H_\theta(t, x(t))$ and $\hat{x}_{1,\theta} = \frac{x(t) - \hat{x}_{0,\theta}}{t}$. By using $v_\theta$ composed of $\hat{x}_{0,\theta}, \hat{x}_{1,\theta}$, EDM enables transportation from $\rho_T = \mathcal{N}(0, T^2 I)$ to $\rho_0$. We apply EDM to our image generation experiments.

**Stochastic Interpolant (SI)**   Stochastic Interpolant (Albergo et al., 2023) facilitates transportation between arbitrary distributions $\rho_0$ and $\rho_1$. The conventional coefficient design is:

$$\alpha(t) = [\alpha_0(t), \alpha_1(t)] = [1 - t, t], \quad T = 1, \quad (4)$$

which represents linear interpolation between $x_0$ and $x_1$. SI models $[\hat{x}_{0,\theta}, \hat{x}_{1,\theta}] = H_\theta(t, x(t))$. Given that SI is useful for transporting between arbitrary distributions, we employ SI for various 2-dimensional experiments to validate our framework.

## 4 METHODOLOGY

### 4.1 DEFINITION OF ADAPTIVE MULTIDIMENSIONAL COEFFICIENT

We introduce the *multidimensional coefficient*, $\gamma(t) = [\gamma_0(t), \gamma_1(t)] \in \mathbb{R}^{2 \times d}$, which generalizes the conventional unidimensional coefficient by extending it to higher dimensions.

**Definition 1** *(**Multidimensional Coefficient**) $\gamma(t) = [\gamma_0(t), \gamma_1(t)] \in \mathbb{R}^{2 \times d}$ for $t \in [0, T]$ defines the trajectory $x(t) = \gamma_0(t) \odot x_0 + \gamma_1(t) \odot x_1$, where $x_0, x_1 \in \mathbb{R}^d$. $\gamma(t)$ must satisfy: $\gamma(t) \in [0, T]$, $\gamma_0(0) = 1_d$, $\gamma_0(T) = k_d$, $\gamma_1(0) = 0_d$, $\gamma_1(T) = T_d$, and $\gamma \in C^1([0, T], \mathbb{R}^{2 \times d})$. Here, $k \in [0, T]$ and $i_d$ denotes a d-dimensional vector filled with the value i.*

$\gamma \in C^1([0, T], \mathbb{R}^{2 \times d})$ indicates that $\gamma$ is continuously first-order differentiable with respect to $t$ on the interval $[0, T]$. Boundary conditions written above ensure that $x(t)$ becomes $x_0$ and $x_T$ for $t = 0$ and $t = T$, which is a requirement for transportation. Values $k$ and $T$ for boundary conditions vary based on the task. For example, in image translation tasks where both distributions $\rho_0$ and $\rho_1$ are data distributions, $k = 0$ and $T = 1$ might be appropriate. The unidimensional coefficient $\alpha$ is a special case of $\gamma$ when all elements $\gamma^{i,j}(t) = \gamma^{i',j'}(t)$ for any indices $i, j, i', j'$ in $\gamma(t)$. We visualize $\alpha$ and $\gamma$ in Figure 1 for better comprehension. The above definition of the multidimensional coefficient uses the same coefficient with respect to the trajectory. However, we can consider a multidimensional coefficient $\gamma$ parameterized by $\phi$, allowing adaptation to different inference trajectories $x_{\theta,\phi}(t)$ for inference times $\tau = \{t_0, \ldots, t_N\}$, where $\theta$ represents the flow or diffusion model parameters:

**Definition 2** *(Adaptive Multidimensional Coefficient)* *For* $t \in [0, T]$ *and inference trajectory* $x_{\theta,\phi}(t)$, *the adaptive multidimensional coefficient* $\gamma_\phi(t, x_{\theta,\phi}(t)) : [0, T] \times \mathbb{R}^d \to \mathbb{R}^{2 \times d}$ *is parameterized by* $\phi$. *Boundary conditions follow Definition 1, with* $\gamma_\phi \in C^1([0, T], \mathbb{R}^{2 \times d})$.

To reduce computational cost in calculating $\gamma_\phi$, we use only $t = T$ for $x_{\theta,\phi}(t)$ in $\gamma_\phi(t, x_{\theta,\phi}(t))$ rather than the inference trajectory at multiple time points. This approach allows us to compute $\gamma_\phi$ across the entire inference time schedule $\tau = \{t_0, \ldots, t_N\}$ with a single function evaluation before initiating transportation. By using the adaptive multidimensional coefficient, we can address multidimensional trajectory optimization as outlined in the next section.

### 4.2 Multidimensional Trajectory Optimization: Definition and Practice

A key aspect of our perspective is that the quality of a trajectory cannot be fully evaluated until the entire transportation process is completed. This contrasts with existing views on trajectory optimality, which pre-define properties for the trajectory without simulation, as seen in works such as Liu et al. (2023); Tong et al. (2024); Singhal et al. (2023); Bartosh et al. (2024). For example, reducing the total trajectory length for transportation, as in the optimal transport (OT) perspective, results in straight trajectories that can indirectly reduce NFE since a straight line minimizes numerical errors when solving differential equations. However, in generative tasks, the real cost is not trajectory length but the NFE required to achieve a certain sample quality. Thus, the optimality for generative models should align more closely with final sample quality, evaluated through simulation, rather than pre-defined properties. Based on this principle, we define trajectory optimality as follows:

**Definition 3** *(Multidimensional Trajectory Optimization (MTO))* *Consider a flow and diffusion-based generator* $G_{\theta,\phi}$ *with fixed configurations (NFE, discretization method, etc.), where* $\theta$ *represents the flow or diffusion model parameters and* $\phi$ *is the parameter for the adaptive multidimensional coefficient* $\gamma_\phi(t, x_T) = [\gamma_{0,\phi}(t, x_T), \gamma_{1,\phi}(t, x_T)] \in \mathbb{R}^{2 \times d}$. *Then the multidimensional trajectory optimization problem is:*

$$\theta^*, \phi^* = \arg\min_{\theta,\phi} \mathbb{D}\left(\rho_1, \hat{\rho}_{1,\theta,\phi}\right), \tag{5}$$

*where* $\hat{\rho}_{1,\theta,\phi}$ *denotes the generated distribution from* $G_{\theta,\phi}$, *and* $\mathbb{D}$ *measures a divergence metric.*

Given that **the trajectory** $x_{\theta,\phi}(t)$ from $G_{\theta,\phi}$ is entirely parameterized by $\theta$ and $\phi$, we have full controllability over the trajectory by adjusting $\theta$ and $\phi$, allowing us to term this process a **trajectory optimization**. This optimization can be approximated $\theta^*, \phi^* \approx \hat{\theta}^*, \hat{\phi}^*$ using a finite set of samples. In this view, we do not guarantee pre-defined properties for the optimized trajectory $x_{\theta^*,\phi^*}(t)$, which underscores our perspective on optimality. Our definition of optimality is based solely on the quality of the final sample for given differential equation-solving configurations, rather than pre-defined properties of the trajectory itself.

**Practical Approach for MTO in High-Dimensional Transportation** To solve MTO in high-dimensional datasets like images using conventional approaches, there are two potential strategies. The first involves simulation-based training, such as CNF (Chen et al., 2019), which is inefficient in terms of both training cost and performance. The second strategy involves the conventional diffusion approach, trained with a fixed single $\phi$, which would require training multiple models $\theta_1, \ldots, \theta_l$ with corresponding coefficients $\phi_1, \ldots, \phi_l$ and then selecting the optimal $\theta$ and $\phi$. This process is computationally intractable. To address these challenges, we propose the following procedure:

1. **Design the hypothesis space of the adaptive multidimensional coefficient** $\gamma_\phi$ **heuristically:** Leverage prior knowledge to identify an appropriate space.

2. **Pre-train flow or diffusion models** $\theta$ **to handle various multidimensional coefficients** sampled from the hypothesis space.

3. **Jointly optimize** $\theta$ **and** $\phi$ **using simulation dynamics and adversarial training:** $\theta$ and $\phi$ in $G_{\theta,\phi}$ aim to deceive discriminator $D_\psi$.

This approach appropriately balances the advantages and disadvantages of simulation-based and simulation-free methods to achieve both trajectory flexibility and training efficiency: employing simulation-based objectives exclusively for $\phi$ after pre-training $\theta$ with various $\gamma$. By following this procedure, we aim to converge to $\theta^*$ and $\phi^*$ that can generate high-quality samples efficiently compared to unoptimized trajectories.

Figure 2: Crude coefficients: (a) Oscillatory behavior in $t$ due to high frequency components; (b) High adjacent pixel differences in $d$. Refined coefficients: (c) Constrained multidimensionality for larger $t$ in pre-training; (d) Unconstrained multidimensionality for adversarial training.

## 4.3 DESIGN CHOICE OF THE COEFFICIENT'S HYPOTHESIS SPACE

Let's define the adaptive multidimensional coefficient space as:

$$\Gamma = \left\{ \gamma : [0, T] \times \mathbb{R}^d \to \mathbb{R}^{2 \times d} \; \middle| \; \begin{array}{l} \gamma = [\gamma_0, \gamma_1], \quad \gamma \in C^1([0, T], \mathbb{R}^{2 \times d}), \\ \gamma_0(0, x_T) = 1_d, \; \gamma_0(T, x_T) = k_d, \\ \gamma_1(0, x_T) = 0_d, \; \gamma_1(T, x_T) = T_d \end{array} \right\}, \tag{6}$$

where $k \in [0, T]$, $T > 0$. To design the hypothesis space $\Gamma_h \subseteq \Gamma$ for $\gamma_\phi$, we consider three main properties. First, the hypothesis space should be broad enough to include the optimal coefficient while avoiding unnecessary complexity to de-burden the flow and diffusion model. As shown in Figure 2, some multidimensional coefficients have excessive high-frequency components in $t$ and across different $d$ dimensions. Given the vast size of the coefficient space, it's crucial to exclude such crude coefficients using appropriate constraints and define a well-designed hypothesis space for $\gamma_\phi$ to explore. Second, the computation of $\gamma_\phi$ by parameter $\phi$ should require low computational cost (NFE). Lastly, for pre-training flow or diffusion models, it should be easy to sample random $\gamma$ from the hypothesis space. Considering these factors, we decide to model the weights of sinusoidals by parameter $\phi$ as in Albergo et al. (2024). Our chosen design is:

$$\Gamma_h = \left\{ \gamma_\phi : [0, T] \times \mathbb{R}^d \to \mathbb{R}^{2 \times d} \; \middle| \; \begin{array}{l} \gamma_\phi = [\gamma_{0,\phi}, \gamma_{1,\phi}], \quad \phi \in \mathcal{P}, \\ \gamma_{0,\phi}(t, x_T) = T \dfrac{f_\phi(t, x_T)}{f_\phi(t, x_T) + g_\phi(t, x_T)}, \\ \gamma_{1,\phi}(t, x_T) = T \dfrac{g_\phi(t, x_T)}{f_\phi(t, x_T) + g_\phi(t, x_T)} \end{array} \right\}, \tag{7}$$

where $\mathcal{P}$ represents the parameter space from which $\phi$ is drawn, and it determines the specific form of the functions $\gamma_{0,\phi}$ and $\gamma_{1,\phi}$ within the space $\Gamma_h$. Above parameterization can vary depending on the flow and diffusion framework. For example, we use $\gamma_{0,\phi}$ as described above for SI, but set $\gamma_{0,\phi}(t, x_T) = 1_d$ for EDM to align with its original formulation. $f_\phi$ and $g_\phi$ are:

$$f_\phi(t, x_T) = 1 - \frac{t}{T} + \left( \sum_{m=1}^{M} w_{m,\phi}^f(x_T) b_m(t) \right)^2, \; g_\phi(t, x_T) = \frac{t}{T} + \left( \sum_{m=1}^{M} w_{m,\phi}^g(x_T) b_m(t) \right)^2, \tag{8}$$

where $b_m(t) = sin(\pi m (t/T)^{1/q}) \in \mathbb{R}$ is sinusoidal with hyperparameter $q$ and $w_\phi(x_1) = [w_\phi^f(x_1), w_\phi^g(x_1)] \in \mathbb{R}^{2 \times M \times d}$ represents the multidimensional weights for the sinusoidals. If $b_m(0) = b_m(T) = 0$, this parameterization always satisfies $\Gamma_h \subseteq \Gamma$. We impose two constraints on $w$: low-pass filtering (LPF) and scaling:

$$w_\phi(x_T) = s \, \text{LPF} \circ \tanh(U_\phi(x_T)), \quad s \in \mathbb{R}, \tag{9}$$

where $U_\phi$ is a U-Net. LPF is implemented by convolution with a Gaussian kernel, applied between different $d$ dimensions, to exclude high frequency in $d$. The scale hyperparameter $s$ adjusts the range of $w_\phi(x_1) \in [-s, s]$. When $s = 0.0$, $\gamma$ reduces to $\alpha$. Details for design are in Appendix A.

This parameterization for $\Gamma_h$ satisfies all three properties mentioned earlier. First, we can exclude high-frequency components in $t$ and $d$ by controlling $s$, $M$, and the configurations of LPF. Second, we can compute $\gamma_\phi$ by modeling the sinusoidal weights with $U_\phi$ as written above, which costs 1 NFE to calculate the entire continuous parameterized coefficient $\gamma_\phi$. Lastly, we can easily sample a random multidimensional coefficient from $\Gamma_h$ by sampling sinusoidal weights from a uniform distribution as $w(u) = s \, \text{LPF} \circ u, \; u \sim \mathcal{N}(-1, 1) \in \mathbb{R}^{2 \times M \times d}$ for pre-training EDM and SI.

**Hypothesis Space for Pre-training and Adversarial Training** Given that a large hypothesis space of coefficients for pre-training can burden $H_\theta$ and potentially degrade performance, we use a smaller hypothesis space for pre-training and open multidimensionality fully across $t$ for adversarial training. Specifically, as shown in Figure 2, we use $\gamma$ with large multidimensionality near $t = 0$ and small multidimensionality for large $t$ by configuring LPF during the pre-training of $H_\theta$. For adversarial training, we fully open multidimensionality for $\gamma_\phi$ across the entire $t$. Further implementation details are provided in Appendix A.

**Coefficient Labeling for Flow and Diffusion** For MTO, it is important for $\gamma_\phi$ to consider the global structure of transportation; hence, $\gamma_\phi$ should receive feedback from flow and diffusion models. This is enabled by incorporating the coefficient information $\gamma$ into $H_\theta(t, x(t), \gamma)$ for pre-training and adversarial training. We concatenate $\gamma$ with $x(t)$ along the channel axis as input to $H_\theta$, enabling coefficient information inclusion without modifying the model structures. The loss function for pre-training EDM and SI is similar to the original except for the additional coeficient label conditioning. Further details are provided in Appendix C.1.

By using the coefficient's hypothesis space and coefficient labeling techniques described above, we train EDM-based diffusion model $H_\theta$ with the objective described simply as below:

$$\mathcal{L}_\theta^{\text{pre}} = \mathbb{E}_{t,x_0,x_1} \| H_\theta(t, x(t), \gamma(t,u)) - x_0 \|_2^2, \; t \sim \mathcal{N}(-1.2, 1.2), \; u \sim \mathcal{N}(-1,1) \in \mathbb{R}^{2 \times M \times d}. \quad (10)$$

By using above loss function, $H_\theta$ is prepared for the trajectory optimization. Detailed version of the loss function for EDM and SI are in Appendix C.1.

### 4.4 ADVERSARIAL APPROACH FOR MULTIDIMENSIONAL TRAJECTORY OPTIMIZATION

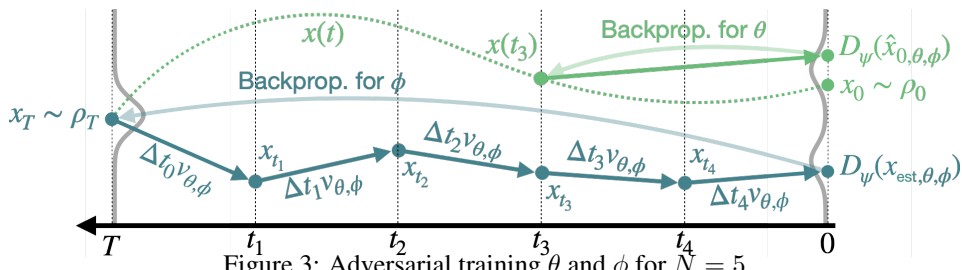

Figure 3: Adversarial training $\theta$ and $\phi$ for $N = 5$.

For EDM, the vector field parameterized by $H_\theta(t, x(t), \gamma_\phi(t, x_T))$ and the coefficient $\gamma_\phi(t, x_T)$ is:

$$v_{\theta,\phi}(t_i, x(t_i), x_T) = \frac{1}{\gamma_{1,\phi}(t_i, x_T)} \odot (x(t_i) - H_\theta(t_i, x(t_i), \gamma_\phi(t_i, x_T))). \quad (11)$$

By using the above vector field, we compose the generator $G(\tau, x_T, v_{\theta,\phi})$ with Euler discretization, where $\tau = \{t_0, \ldots, t_N\}$ with $t_0 = T > \ldots > t_N = 0$ represents the inference time schedule (details are in Appendix B). Following Goodfellow et al. (2014), we can minimize Equation 5 by solving the following min-max problem:

$$\min_{\theta,\phi} \max_\psi \mathbb{E}_{x_0} [\log D_\psi(x_0)] + \mathbb{E}_{x_T} [\log (1 - D(G(\tau, x_T, v_{\theta,\phi})))], \quad (12)$$

where $\psi$ represents the discriminator parameter. As shown, $\theta$ and $\phi$ in $G_{\theta,\phi}$ aim to deceive $D_\psi$. Specifically, for training stability and better performance, we employ the StyleGAN-XL (Sauer et al., 2022) discriminator for $D_\psi$ with hinge loss (Lim & Ye, 2017), as used in Kim et al. (2024). The key point is that we use different loss functions for $\theta$ and $\phi$: we only use the simulation-based objective for $\phi$, as shown in Figure 3. The hinge loss functions for $\phi$ and $\psi$ are:

$$\begin{aligned}
\mathcal{L}_\phi &= -\mathbb{E}_{x_T \sim \rho_T}[D_\psi(G(\tau, x_T, v_{\theta,\phi}))], \\
\mathcal{L}_\psi &= \mathbb{E}_{x_0 \sim \rho_0}[\max(0, 1 - D_\psi(x_0))] + \mathbb{E}_{x_T \sim \rho_1}[\max(0, 1 + D_\psi(G(\tau, x_T, v_{\theta,\phi})))],
\end{aligned} \quad (13)$$

where $\mathcal{L}_\phi$ and $\mathcal{L}_\psi$ indicate that gradients are calculated with respect to $\phi$ and $\psi$. We also optimize $\theta$ using the adversarial objective from $D_\psi$ as follows:

$$\mathcal{L}_\theta = -\mathbb{E}_{x_1 \sim \rho_1}[D_\psi(H_\theta(t, x(t), \gamma_\phi(t, z)))], \quad x(t) = x_0 + \gamma_{1,\phi}(t, z) \odot x_1, \quad z \sim \rho_T, \quad (14)$$

where $\mathcal{L}_\theta$ indicates that the gradient is calculated only with respect to $\theta$. Since $\theta$ only needs to handle elements in $\{\gamma_\phi(t, x_T) \mid t \in [0, T], x_T \sim \rho_T\}$ and not other elements in $\Gamma_h$, $\theta$ is trained exclusively on $\gamma_\phi$, reducing the load on $H_\theta$. With these loss functions, $\phi$ is optimized to find better trajectories, while $\theta$ adapts to $\gamma_\phi$, which is sparser than $\Gamma_h$. The final loss term for MTO is:

$$\mathcal{L}^{\text{MTO}}_{\theta,\phi,\psi} = \mathcal{L}_\theta + \mathcal{L}_\phi + \mathcal{L}_\psi. \tag{15}$$

The algorithm is summarized in the following pseudo-code:

---

**Algorithm 1** Pre-training EDM-based $H_\theta$ and Adversarial Training for MTO

---

1: **Input** $G(\tau, v_{\theta,\phi}(H_\theta, \gamma_\phi)), D_\psi$
2: **while** True **do**                     ▷ Pre-training $H_\theta$ with randomly sampled trajectories
3:    Sample $t \sim \mathcal{N}(-1.2, 1.2)$, $x_0 \sim \rho_0$, $x_1 \sim \rho_1$, and $u \sim \mathcal{U}(-1, 1)$
4:    $\mathcal{L}^{\text{pre}}_\theta = \mathbb{E}_{x_0,x_1} \| H_\theta\left(t, x(t), \gamma(t, u)\right) - x_0 \|^2_2$; Update $\theta$
5:    **if** $\theta$ converges **then break**
6:    **end if**
7: **end while**
8: **while** True **do**                     ▷ Training $H_\theta$ and $\gamma_\phi$ via adversarial training
9:    Sample $t \sim \mathcal{N}(-1.2, 1.2)$, $x_0 \sim \rho_0$, and $x_T, z \sim \rho_T$
10:    $\hat{x}_{0,\theta,\phi} \leftarrow H_\theta(t, x(t), \gamma_\phi(t, z))$
11:    $x_{\text{est},\theta,\phi} \leftarrow G(\tau, x_T, v_{\theta,\phi})$
12:    $\mathcal{L}^{\text{MTO}}_{\theta,\phi,\psi}$; Update $\theta, \phi, \psi$
13:    **if** $\theta, \phi, \psi$ converge **then break**
14:    **end if**
15: **end while**

---

## 5 EXPERIMENTS

### 5.1 2-DIMENSIONAL TRANSPORTATION

We conduct experiments on 2-dimensional synthetic datasets provided by Tong et al. (2024), using the Stochastic Interpolants framework. As these experiments aim to validate the existence of performance gains achievable through optimizing $\phi$, **we solely train $\phi$ while freezing $\theta$** for a fair comparison with baseline methods. Additionally, since calculating the 2-Wasserstein distance $\mathcal{W}_2$ as the divergence term in Equation 5 is feasible for a 2-dimensional dataset, we use the loss function $\mathcal{L}_\phi = \mathcal{W}_2(x_0, x_{\text{est},\theta,\phi})$. To validate that using an adaptive trajectory from MTO offers better transportation even where optimality is defined by a straight trajectory, we experiment with additional configurations for minibatch pairing $(x_0, x_1)$: random pairing and OT pairing (Tong et al., 2024). The minibatch-OT method encourages the flow and diffusion model to learn a straight trajectory by pairing $x_0$ and $x_1$ as OT within a minibatch during training, where optimality for the trajectory is defined as straight. Detailed training configurations are presented in Appendix C.

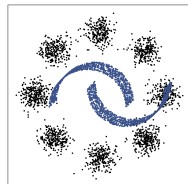

Figure 4: $x_T$ (black) and $x_0$ (blue).

Table 1: $\mathcal{W}_2$ distance $\downarrow$ for 2-dimensional transportation results.

| | Gaussian to 8 Gaussians | | Gaussian to Moons | | 8 Gaussians to Moons | | Moons to 8 Gaussians | |
|---|---|---|---|---|---|---|---|---|
| NFE → | 5 | 10 | 5 | 10 | 5 | 10 | 5 | 10 |
| SI | $0.763_{\pm0.040}$ | $0.673_{\pm0.055}$ | $0.882_{\pm0.035}$ | $0.643_{\pm0.060}$ | $0.981_{\pm0.112}$ | $0.649_{\pm0.165}$ | $1.271_{\pm0.185}$ | $0.998_{\pm0.203}$ |
| SI$_{\text{MTO}}$ | $\mathbf{0.721}_{\pm0.082}$ | $\mathbf{0.452}_{\pm0.033}$ | $\mathbf{0.682}_{\pm0.093}$ | $\mathbf{0.359}_{\pm0.098}$ | $\mathbf{0.924}_{\pm0.235}$ | $\mathbf{0.311}_{\pm0.051}$ | $\mathbf{0.908}_{\pm0.109}$ | $\mathbf{0.500}_{\pm0.072}$ |
| OT-SI | $0.457_{\pm0.021}$ | $0.440_{\pm0.052}$ | $0.245_{\pm0.023}$ | $0.217_{\pm0.019}$ | $0.321_{\pm0.064}$ | $0.318_{\pm0.068}$ | $0.488_{\pm0.050}$ | $0.492_{\pm0.056}$ |
| OT-SI$_{\text{MTO}}$ | $\mathbf{0.399}_{\pm0.017}$ | $\mathbf{0.415}_{\pm0.016}$ | $\mathbf{0.230}_{\pm0.015}$ | $\mathbf{0.188}_{\pm0.006}$ | $\mathbf{0.258}_{\pm0.015}$ | $\mathbf{0.221}_{\pm0.014}$ | $\mathbf{0.421}_{\pm0.012}$ | $\mathbf{0.407}_{\pm0.031}$ |

As shown in Table 1, MTO consistently achieves the best results, even for models trained with minibatch-OT. This suggests that a straight trajectory is not always optimal even in OT-trained model, and MTO can adaptively discover better trajectories to correct errors that arise during transportation. Figure 5 further illustrates how MTO adjusts the trajectory direction to optimize transportation, resulting in a path that is not straight. A comparison of (c) with (d) reveals a distinct piece-wise linear trajectory in (d), indicating that MTO's trajectory isn't straight but achieves superior performance.

One critical source of error in transportation arises from the simulation-free dynamic objectives. In these objectives, pre-defining the trajectory forces the model to follow it, but achieving perfect consistency in trajectory simulation is challenging, even in the minibatch-OT setting,

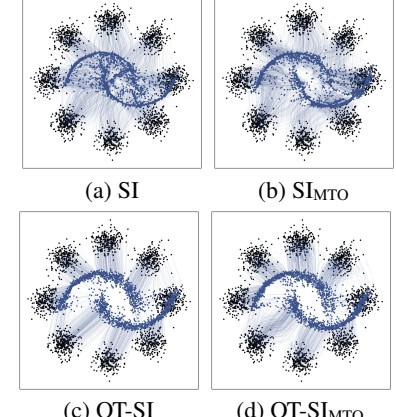

(a) SI          (b) SI$_{\text{MTO}}$

(c) OT-SI       (d) OT-SI$_{\text{MTO}}$

Figure 5: Comparison of inference trajectories from 8 Gaussians to Moons.

leading to noisy training and approximation limitations. Additionally, even with consistent trajectory supervision, errors inevitably emerge during actual trajectory simulations due to the inherent imperfections in model training. MTO addresses these issues by leveraging simulation dynamics to adaptively find an optimal trajectory for each $x_1$ within the given $\theta$ and differential equation solver configurations. These results empirically suggest that the optimality of the trajectory, in terms of transportation quality, is not necessarily determined by the pre-defined property of the trajectory, which contrasts with conventional perspectives.

## 5.2 IMAGE GENERATION

Table 2: Performance comparisons on CIFAR-10.

| Model | NFE | Unconditional | | Conditional |
| | | FID↓ | IS↑ | FID↓ |
| --- | --- | --- | --- | --- |
| **GAN Models** | | | | |
| StyleGAN-Ada (Karras et al., 2020) | 1 | 2.92 | **9.82** | 2.42 |
| StyleGAN-XL (Sauer et al., 2022) | 1 | – | – | **1.85** |
| StyleSAN-XL (Takida et al., 2024) | 1 | **1.36** | – | – |
| **Diffusion Models** | | | | |
| DDPM (Ho et al., 2020) | 1000 | 3.17 | **3.75** | – |
| DDIM (Song et al., 2022) | 100 | 4.16 | – | – |
| Score SDE (Song et al., 2021) | 2000 | 2.20 | 3.45 | – |
| EDM (Karras et al., 2022) | 35 | **1.97** | - | 1.79 |
| **Diffusion Models – Distillation** | | | | |
| KD (Luhman & Luhman, 2021) | 1 | 9.36 | – | – |
| PD (Salimans & Ho, 2022) | 1 | 9.12 | – | – |
| PD (Salimans & Ho, 2022) | 2 | 4.51 | – | – |
| DFNO (Zheng et al., 2023) | 1 | 5.92 | – | – |
| 2-Rectified Flow (Liu et al., 2023) | 1 | 4.85 | – | – |
| CD (Song et al., 2023) (Song et al., 2023) | 1 | 3.55 | – | – |
| CD (Song et al., 2023) | 2 | 2.93 | – | – |
| CD + GAN (Lu et al., 2023) | 1 | 2.65 | – | – |
| GDD (Zheng & Yang, 2024) | 1 | 1.66 | **10.11** | 1.58 |
| GDD-I (Zheng & Yang, 2024) | 1 | **1.54** | 10.10 | **1.44** |
| CTM (Kim et al., 2024) | 1 | 1.98 | – | 1.73 |
| CTM (Kim et al., 2024) | 2 | 1.87 | – | 1.63 |
| CTM (Kim et al., 2024) | 5 | 1.86 | – | 1.98 |
| CTM (Kim et al., 2024) | 6 | 1.93 | – | 2.04 |
| **Diffusion Models - MTO** | | | | |
| EDM-MTO (ours) | 5 (+) | **1.69** | 9.43 | 1.37 |

Table 3: Performance comparisons on FFHQ-64x64.

| Model | NFE | FID ↓ |
| --- | --- | --- |
| DiffusionGAN (Wang et al., 2023) | 1 | **2.83** |
| **Diffusion Models** | | |
| EDM (Karras et al., 2022) | 79 | **1.96** |
| **Diffusion Models - Distillation** | | |
| SiD (Zhou et al., 2024) | 1 | 1.71 |
| SiD (Zhou et al., 2024) | 1 | 1.55 |
| GDD (Zheng & Yang, 2024) | 1 | 1.08 |
| GDD-I (Zheng & Yang, 2024) | 1 | **0.85** |
| **Diffusion Models - MTO** | | |
| EDM-MTO (ours) | 5 (+) | **2.27** |

Table 4: Performance comparisons on AFHQv2-64x64.

| Model | NFE | FID ↓ |
| --- | --- | --- |
| **Diffusion Models** | | |
| EDM (Karras et al., 2022) | 79 | **2.39** |
| **Diffusion Models - Distillation** | | |
| SiD (Zhou et al., 2024) | 1 | 1.62 |
| GDD (Zheng & Yang, 2024) | 1 | **1.23** |
| GDD-I (Zheng & Yang, 2024) | 1 | 1.31 |
| **Diffusion Models - MTO** | | |
| EDM-MTO (ours) | 5 (+) | **2.04** |

We apply adversarial approach for MTO to CIFAR-10 (Krizhevsky & Hinton, 2009), FFHQ (Karras et al., 2018), and AFHQv2 (Choi et al., 2020) datasets with $N = 5$. We utilize the EDM-VP training configurations for $H_\theta$, the U-Net architecture from Song et al. (2021) for $U_\phi$, and the StyleGAN-XL (Sauer et al., 2022) discriminator for $D_\psi$. $U_\phi$ also incorporates labels for CIFAR-10 conditional generation. We measure Fréchet Inception Distance (FID) (Heusel et al., 2017) and Inception Score (IS) (Salimans et al., 2016). Detailed configurations are available in Appendix C.

Table 5: FID ↓ for ablation study on CIFAR-10. -$\alpha$ and -$\gamma$ denote coefficients used for pre-training.

| Configuration ↓ \ NFE → | Unconditional | | Conditional | |
| | 5 (Euler) | 35 (Heun) | 5 (Euler) | 35 (Heun) |
| --- | --- | --- | --- | --- |
| EDM-$\alpha$ | 68.73 | 1.97 | 48.76 | 1.79 |
| EDM-$\gamma$ | 69.58 | 2.08 | 48.53 | 1.81 |
| EDM-$\gamma$ + Adv. $\phi$ (no multi.) | 33.55 | – | 25.56 | – |
| EDM-$\gamma$ + Adv. $\phi$ | 18.67 | – | 7.77 | – |
| EDM-$\gamma$ + Adv. $\theta$ | 2.28 | – | 2.14 | – |
| EDM-$\gamma$ + Adv. $\theta, \phi$ | **1.81** | – | **1.42** | – |

By appropriately constraining $\gamma(t, u)$ during pre-training of $H_\theta$, we nearly maintain $H_\theta$'s performance despite the increased complexity compared to training with $\alpha$, as demonstrated in Table 5.

**Impact of MTO on Image Generation** As shown in Tables 2, 3, and 4, our approach generates high-quality samples across various datasets with only 5 (+) NFE (+ for the calculation of $\gamma_\phi$, given that the network for $\gamma_\phi$ is smaller than the network for $H_\theta$), reaching a state-of-the-art result (FID = 1.37) on CIFAR-10 conditional generation. Except for FFHQ, EDM-MTO achieves better performance than EDM with fewer NFE. For a fair comparison with other distillation methods in terms of NFE, we select CTM (Kim et al., 2024) as a representative due to its popularity, high performance, and the use of the same model architecture based on EDM and adversarial training. We

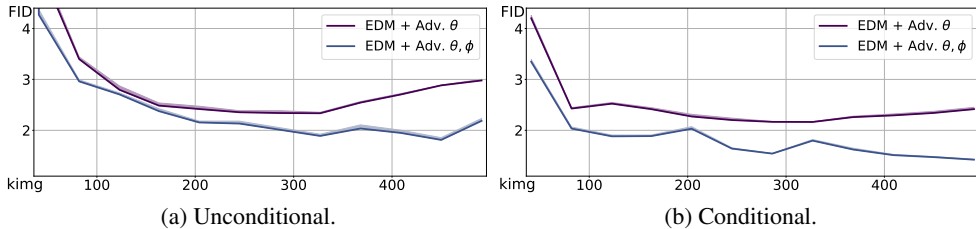

(a) Unconditional.  (b) Conditional.

Figure 6: EDM-$\gamma$ + Adv. $\theta$ and EDM-$\gamma$ + Adv. $\theta, \phi$.

then calculate the FID using 5 and 6 NFE. As shown in Table 2, increasing the NFE of CTM does not significantly decrease the FID, and the FID even increases. This indicates that the adversarial approach for MTO provides additional performance gains that cannot be achieved by distillation methods alone, even with increased computational cost.

To empirically validate trajectory optimization's benefit for high-dimensional generation, we conduct ablation studies by training either $\theta$ or $\phi$ individually, similar to the 2-dimensional experiments in Section 5.1. As presented in Table 5, the results reveal that jointly training $\theta$ and $\phi$ yields the best performance. Figure 6 further illustrates that FID decreases more significantly during joint training

Table 6: Comparison of required kimg for training.

| Dataset | GDD-I | Ours |
|---------|-------|------|
| CIFAR-10 | 5000 | 1382 |
| FFHQ | 10000 | 106 |
| AFHQv2 | 10000 | 955 |

compared to training $\theta$ alone. Interestingly, training only $\phi$ also significantly reduces FID (18.67, 7.77) compared to EDM-$\gamma$. These findings suggest that MTO's performance improvements stem not only from the adversarial training of $H_\theta$ but also from the combined training of both $H_\theta$ and $\gamma_\phi$, indicating the existence of performance gains achievable only through MTO.

**Training Efficiency and Scalability of Our Approach** The adversarial approach for MTO demonstrates remarkable training efficiency despite incorporating simulation-based training. As shown in Table 6, the required number of training images for our approach is lower across all datasets compared to GDD-I (Zheng & Yang, 2024), a method known for its efficiency in diffusion distillation. Additionally, FID significantly decreases in the early stages of training, as illustrated in Figure 6. Training times for the adversarial approach are 10, 2, and 6 hours for CIFAR-10, FFHQ, and AFHQv2, respectively, which are also comparatively low. The primary cost of simulation dynamics arises from VRAM requirements. However, training remains practically feasible, as good performance can be achieved with just 5 NFE, which is relatively low. All our experiments for adversarial training were conducted on GPUs with 48GB of VRAM-less than the 80GB VRAM GPUs frequently used in related works. These results not only highlight the effectiveness of simulation-based end-to-end optimality but also showcase the strength of combining simulation-free and simulation-based methodologies, leveraging the advantages of each while mitigating their limitations. Considering these aspects, we estimate that our adversarial approach for MTO is scalable to larger datasets while maintaining efficiency.

**Impact of Multidimensionality for MTO** To examine how multidimensionality influences performance, we train $\phi$ with different configurations by averaging $\tanh(U_\phi)$ across specific axes. For example, to retain multidimensionality solely in the height dimension ([F, T, F]), we use the same $w_\phi$ in the channel and width dimensions by taking mean in those axes. As shown in Figure 7, incorporating more axes consistently leads to performance improvements, indicating that trajectory multidimensionality positively impacts generation quality.

**Analysis of Trained Sinusoidal Weights** To visualize the trained $w_\phi$, we plot t-SNE embeddings of four different weights across all datasets, as shown in Figure 9. Notably, $w_\phi$ diverges from weights randomly sampled from the pre-defined hypothesis space and is far from unidimensional coefficients. This suggests that during joint adversarial training of $\theta$ and $\phi$, $\gamma_\phi$ adaptively identifies optimal coefficients with-

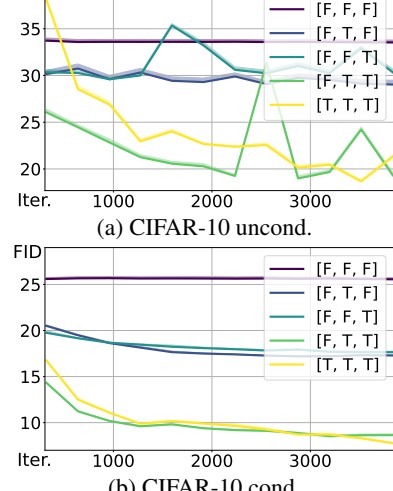

(a) CIFAR-10 uncond.

(b) CIFAR-10 cond.

Figure 7: EDM-$\gamma$ + Adv. $\phi$.

out heavily depending on the pre-trained distribution of $\gamma(t, u)$. Interestingly, $\gamma_\phi$ exhibits a sparser distribution in CIFAR-10 conditional generation than in the unconditional setting, showing similar $w_\phi$ values for the same label condition. This suggests that the optimality of the coefficient depends

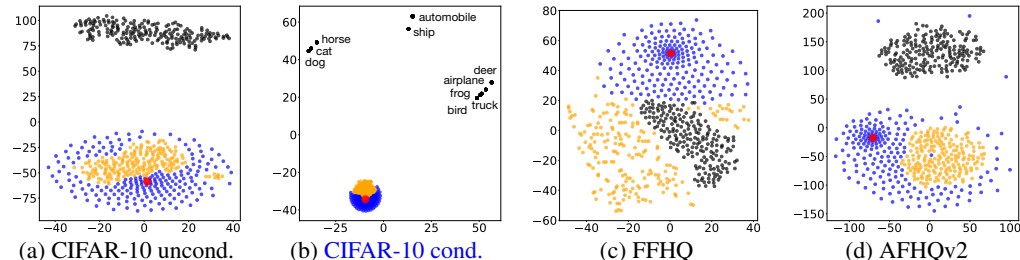

Figure 9: T-SNE for various coefficients. Red: $w = 0$ (unidimensional coefficient), Blue: $w = su$, Orange: $w = s$ LPF $\circ u$ (for pre-training), Black: $w = w_\phi$ (trained).

more on the label condition than on the starting point ($x_T$) of the differential equation. This sparse $w_\phi$ distribution may contribute to the high performance (SOTA) and training stability (as shown in Figure 7) observed in the adversarial approach to MTO for conditional generation. When $\gamma_\phi$'s output is less varied, $H_\theta$ has a reduced learning burden for diverse paths during adversarial training, potentially enhancing performance. These findings indicate that the adversarial approach to MTO can be particularly effective in conditional generation settings compared to unconditional generation.

Additionally, to validate that the optimized trajectory is not straight, we calculate the $L_2$ norm of the difference between a straight trajectory $x_t = \frac{t}{T}x_1 + (1 - \frac{t}{T})x_{\text{est},\theta,\phi}$ and the optimized inference trajectory $x_{\theta,\phi}(t)$. As shown in Figure 8, the optimized trajectory deviates from the straight trajectory. These findings demonstrate that the adversarial approach for MTO effectively discovers superior, non-linear trajectories in high-dimensional datasets, thereby enhancing overall performance. Additional experiments for various empirical performance validation are in Appendix E.

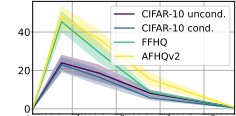

Figure 8: The difference between the straight and the optimized trajectory.

## 6 CONCLUSIONS

In this work, we extend the conventional use of unidimensional coefficients in flow and diffusion models by introducing adaptive multidimensional coefficients. We present the problem of Multidimensional Trajectory Optimization, which aims to identify adaptive trajectories that improve generative performance under fixed solver configurations and a specified starting point of the differential equation. This approach introduces a different perspective on trajectory optimality, focusing on the quality of the final transportation outcome rather than pre-defined properties of the trajectory, such as straightness. Our proposed solution pre-trains flow and diffusion models with various coefficients to prepare for MTO, and utilizes simulation dynamics combined with adversarial training to perform MTO. This approach effectively learns multidimensional trajectories, as validated through experiments across various generative tasks and datasets. These experiments demonstrate that our method identifies more efficient trajectories, leading to significant performance improvements in transportation tasks. Importantly, this work achieves full trajectory flexibility and adaptability through end-to-end adversarial training—previously only achievable with the high training costs of simulation-based objectives—while preserving training efficiency. By enhancing the performance of flow and diffusion models, we hope this work inspires further exploration and advancements in this field.

## 7 LIMITATIONS AND FUTURE WORKS

First, since we use coefficient labeling (Section 4.3) for the diffusion model, the model structure differs from existing pre-trained flow and diffusion models. As a result, training models from scratch is required, which can be cumbersome. This issue could be mitigated in future works by replacing a few layers from well pre-trained model and using it as initialization for pre-training. Second, $\gamma_\phi$ is tied to the specific sampling configuration used during MTO, limiting its flexibility for inference under alternative configurations. Future work could address this by conditioning $\gamma_\phi$ on diverse sampling configurations, enabling better adaptability and efficiency. Third, refining the design of $\gamma_\phi$ could improve efficiency. As shown in Table 3 and Table 4, our method's FID is higher than distillation methods for larger datasets, potentially due to using the same model size and NFE configurations as smaller datasets like CIFAR-10. Optimizing $\gamma_\phi$ for larger datasets could reduce model size and NFE requirements while maintaining performance. Lastly, while MTO empirically demonstrates improved performance across datasets, its theoretical foundation remains unexplored. A potential connection lies with Latent Diffusion Models (LDM) (Rombach et al., 2022), where MTO's adaptive trajectories resemble the space warping in LDM. Unlike LDM, which compresses latent space, MTO achieves warping without altering dimensionality, offering a novel perspective on trajectory optimization. We hope these limitations inspire further research in this area.

REPRODUCIBLILITY STATEMENT

We will release the codes upon paper acceptance.

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

## A    HYPOTHESIS SPACE DESIGN FOR MULTIDIMENSIONAL COEFFICIENT

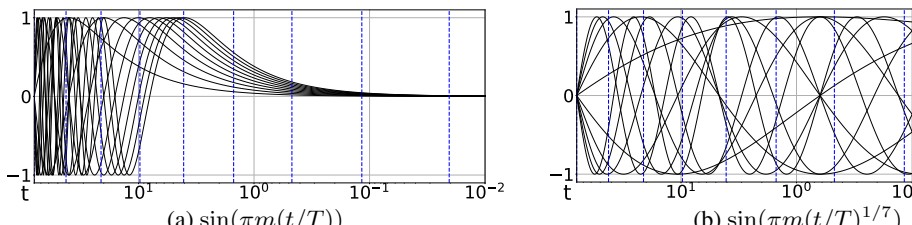

(a) $\sin(\pi m(t/T))$          (b) $\sin(\pi m(t/T)^{1/7})$

Figure 10: Comparison of $b_m(t)$ for $M = 10$. The blue dotted line represents the EDM inference time schedule for $N = 10$.

**Design Choice of Sinusoidals**    The inference time schedule of EDM is defined as:

$$t_i = \left( t_{\max}^{\frac{1}{q}} + \frac{i}{N-1} \left( t_{\min}^{\frac{1}{q}} - t_{\max}^{\frac{1}{q}} \right) \right)^q, \quad t_{\min} = 0.002, \ t_{\max} = 80, \ q = 7. \tag{16}$$

As illustrated in Figure 10, $b_m(t) = \sin(\pi m(t/T)^{1/7})$ effectively covers the entire EDM time schedule, while $b_m(t) = \sin(\pi m(t/T))$ does not have much value in $t \leq 1$. Since $w_\phi$ is constrained to $[-s, s]$, this choice can significantly affect $\gamma_\phi$'s controllability during simulation. We use $q = 1$ for SI and $q = 7$ for EDM.

For $b_m(t) = \sin(\pi m(t/T))$, we can consider the aliasing effect, where the sampling rate $f_s = N \geq 2f_{\max} = M$ must hold for choosing an appropriate $M$, ensuring sufficient frequency resolution. Following this principle, we set $M = N$ for all of our adversarial training to avoid aliasing and ensure accurate trajectory optimization.

**Low-Pass Filtering (LPF)**    For low-pass filtering, we apply 2D convolution with a Gaussian kernel where the kernel size is $\frac{20 \times \text{resolution}}{32} - 1$ and Gaussian kernel's $\sigma = \frac{4.0 \times \text{resolution}}{32}$, with resolution referring to the image height or width. To remove boundary effects caused by LPF, we apply zero padding of $\frac{\text{kernel size}+1}{2}$ on all sides of $U_\phi$'s input and crop the edge after LPF to match the input shape. We ensure consistent scaling by calculating the min and max values before LPF per batch and rescaling each batch post-LPF to match the original scale.

**Hypothesis Space for Pre-training and Adversarial Training**    Given that a large hypothesis space of coefficients for pre-training can burden $H_\theta$ and potentially degrade performance, we use a smaller hypothesis space for pre-training and open multidimensionality across $t$ for adversarial training. Specifically, we set the convolution group size for LPF as 1, which makes the output of LPF have the shape $[B, 1, \text{resolution}, \text{resolution}]$. This constrains $\gamma$ to have multidimensionality for small $t$ and reduced multidimensionality for large $t$. For adversarial training, we set the convolution group size for LPF as $[B, 2 \times 3 \times M, \text{resolution}, \text{resolution}]$, resulting in the output channel shape $[B, 2 \times 3 \times M, \text{resolution}, \text{resolution}]$.

## B    DETAILS FOR FLOW-BASED GENERATOR

The displacement of the trajectory $x(t_{i+1}) - x(t_i)$, parameterized by $v_{\theta,\phi}$, is expressed as:

$$\Delta t_i v_{\theta,\phi}(t_i, x(t_i), x_T) = \Delta t_i \dot{\gamma}_{0,\phi}(t_i, x_T) \odot \hat{x}_{0,\theta} + \Delta t_i \dot{\gamma}_{1,\phi}(t_i, x_T) \odot \hat{x}_{1,\theta}$$
$$\approx \Delta \gamma_{0,\phi}(t_i, x_T) \odot \hat{x}_{0,\theta} + \Delta \gamma_{1,\phi}(t_i, x_T) \odot \hat{x}_{1,\theta}, \tag{17}$$

where the time displacement $\Delta t_i = t_{i+1} - t_i$ is from the inference time schedule $\tau = \{t_0, \ldots, t_N\}$ with $t_0 = T > \ldots > t_N = 0$ and $N$ is the NFE. The trajectory displacements are:

$$\Delta \gamma_{0,\phi}(t_i, x_T) = \gamma_{0,\phi}(t_{i+1}, x_T) - \gamma_{0,\phi}(t_i, x_T), \quad \Delta \gamma_{1,\phi}(t_i, x_T) = \gamma_{1,\phi}(t_{i+1}, x_T) - \gamma_{1,\phi}(t_i, x_T) \tag{18}$$

This approach reduces numerical errors when solving differential equations for curved $\gamma$. For EDM, the displacement of the trajectory can be written as:

$$\Delta t_i v_{\theta,\phi}(t_i, x(t_i), x_T) = \frac{\Delta \gamma_{1,\phi}(t_i, x_T)}{\gamma_{1,\phi}(t_i, x_T)} \odot \left( x(t_i) - H_\theta(t_i, x(t_i), \gamma_\phi(t_i, x_T)) \right). \tag{19}$$

For SI:

$$\Delta t_i v_{\theta,\phi}(t_i, x(t_i), x_T) = \Delta\gamma_{0,\phi}(t_i, x_T) \odot H_{0,\theta} + \Delta\gamma_{1,\phi}(t_i, x_T) \odot H_{1,\theta}, \tag{20}$$

where $[x_0, x_T] \approx [\hat{x}_{0,\theta}, \hat{x}_{1,\theta}] = [H_{0,\theta}(t_i, x(t_i), \gamma_\phi(t_i, x_T)), H_{1,\theta}(t_i, x(t_i), \gamma_\phi(t_i, x_T))]$.

The generator $G$ using Euler discretization is then defined as:

$$G(\tau, x_T, v_{\theta,\phi}) = x_{\text{est},\theta,\phi} = x_T + \sum_{i=0}^{N-1} \Delta t_i v_{\theta,\phi}(t_i, x_{\theta,\phi}(t_i), x_T),$$

$$x_{\theta,\phi}(t_{i+1}) \leftarrow x_{\theta,\phi}(t_i) + \Delta t_i v_{\theta,\phi}(t_i, x_{\theta,\phi}(t_i), x_T) \tag{21}$$

## C  DETAILS FOR TRAINING

### C.1  PRE-TRAINING EDM AND SI

Table 7: Hyperparameters used for pre-training EDM.

| Hyperparameter | CIFAR-10 | FFHQ & AFHQv2 |
|---|---|---|
| Number of GPUs | 8 | 8 |
| Duration (Mimg) | 200 | 200 |
| Minibatch size | 512 | 256 |
| Learning rate | 1e-3 | 2e-4 |
| LR ramp-up (Mimg) | 10 | 10 |
| EMA half-life (Mimg) | 0.5 | 0.5 |
| Dropout probability | 13% | 5% (FFHQ) / 25% (AFHQv2) |
| Channel multiplier | 128 | 128 |
| Channels per resolution | 2-2-2 | 1-2-2-2 |
| Augment probability | 12% | 15% |
| $M$ | 10 | 10 |
| Low-pass filtering | True | True |
| $s$ | 0.05 | 0.05 |

**EDM**  We use the code provided by Karras et al. (2022) and follow EDM's configuration, except replacing the unidimensional coefficient $\alpha$ with the multidimensional coefficient $\gamma$:

$$\mathcal{L}_\theta = \mathbb{E}_{t,x_0,x_T} \left[ \lambda(t) c_{\text{out}}(t)^2 \| F_\theta(c_{\text{noise}}(t), c_{\text{in}}(t)x(t), c_{\text{traj}}(t)) - \frac{1}{c_{\text{out}}(t)}(x_0 - c_{\text{skip}}(t)x(t)) \|_2^2 \right], \tag{22}$$

where:

$$c_{\text{in}}(t) = \frac{1}{\sqrt{\gamma_0^2(t, u) + \sigma_{\text{data}}^2}},$$

$$c_{\text{out}}(t) = \frac{\gamma_0(t, u) \cdot \sigma_{\text{data}}}{\sqrt{\sigma_{\text{data}}^2 + \gamma_0^2(t, u)}},$$

$$c_{\text{skip}}(t) = \frac{\sigma_{\text{data}}^2}{\gamma_0^2(t, u) + \sigma_{\text{data}}^2}, \tag{23}$$

$$c_{\text{noise}}(t) = \frac{1}{4} \ln t,$$

$$c_{\text{traj}}(t) = \frac{1}{4} \ln \gamma_0(t, u),$$

$$\lambda(t) = \frac{\gamma_0^2(t, u) + \sigma_{\text{data}}^2}{(\gamma_0(t, u) \cdot \sigma_{\text{data}})^2},$$

where $t$ is sampled from $\ln(t) \sim \mathcal{N}(-1.2, 1.2^2)$ and $u \sim \mathcal{N}(-1, 1) \in \mathbb{R}^{2 \times M \times d}$. $\sigma_{\text{data}} = 0.5$. Both $c_{\text{in}}(t)x(t)$ and $c_{\text{traj}}$ are $d$-dimensional vectors, so we concatenate $[c_{\text{in}}(t)x(t), c_{\text{traj}}]$ as the U-Net input. We used the Adam optimizer with $\beta_1, \beta_2 = [0.9, 0.999]$ and $\epsilon = 1e - 8$.

**SI** We follow the code provided by Tong et al. (2024), using an MLP consisting of 4 linear layers with 64 hidden units and SiLU activation functions. We train SI with a batch size of 256 and 20,000 iterations. loss function for SI is:

$$\mathcal{L}_k(\theta) = \int_0^1 \mathbb{E}[|H_{k,\theta}(t, x(t), \gamma(t, u))|^2 - 2x_k \cdot H_{k,\theta}(t, x(t), \gamma(t, u))]dt, \quad k = 0, 1, \tag{24}$$

## C.2 MULTIDIMENSIONAL TRAJECTORY OPTIMIZATION

Table 8: Hyperparameters used for adversarial training.

| Hyperparameter | CIFAR-10 | FFHQ & AFHQv2 |
|---|---|---|
| Number of GPUs | 8 | 8 |
| Duration for $D_\psi$ (kimg) | 1500 | 1000 |
| Minibatch size for $v_\theta$ | 512 | 256 |
| Minibatch size for $\gamma_\phi$ | 128 | 64 |
| Learning rate for $v_\theta$ | 1e-5 | 1e-5 |
| Learning rate for $\gamma_\phi$ | 1e-4 | 1e-4 |
| Learning rate for $D_\psi$ | 1e-3 | 1e-3 |
| EMA half-life (kimg) | 10 | 10 |
| $M$ | 5 | 5 |
| Low-pass filtering | True | True |
| $s$ | 0.05 | 0.05 |

**EDM** For $U_\phi$, we utilize a U-Net architecture based on Song et al. (2021) with the following settings: 256 channels, [1, 2, 4] channel multipliers, a dimensionality multiplier of 4, 4 blocks, and an attention resolution of 16. The embedding layer for $t$ is disabled. Both $H_\theta$ and $\gamma_\phi$ are made deterministic by disabling dropout. We employ the Adam optimizer with $\beta_1, \beta_2 = [0.0, 0.99]$ and $\epsilon = 1e - 8$. For training $\theta$, we sample $\ln(t) \sim \mathcal{N}(-1.2, 1.2^2)$ and quantize it according to the inference time schedule $\tau$. For ablation studies, each configuration is trained for 500 kimg, which is approximately 4000 iterations. When training $\phi$ independently, LPF is not applied.

**SI** For training $\gamma_\phi$, we use a batch size of 1024 with 2000 iterations, with $s = 0.1$. We don't use low-pass filtering for 2-dimensional experiments and find that training $\phi$ alone is sufficient. Each configuration is trained 3 times, and the mean and standard deviation of the Wasserstein distance are reported.

All experiments are conducted on RTX 4090 Ti and RTX 6000 Ada GPUs.

## D METRICS CALCULATION

For Fréchet Inception Distance (FID) calculation, we follow the code provided by Karras et al. (2022), using 50,000 generated images. We calculate FID three times for each experiment and report the minimum value. The inception score is calculated using the torchvision library.

## E ADDITIONAL EXPERIMENTS

To further validate MTO's empirical benefits, we perform MTO using various flow and diffusion methodologies (SI (Stochastic Interpolants), FM (Flow Matching), and DDPM (Denoising Diffusion Probabilistic Model)) on image datasets (CIFAR-10, ImageNet-32).

## E.1 NETWORK ARCHITECTURES

Table 9: U-Net configurations for $H_\theta$.

| Configuration | CIFAR-10 | ImageNet-32 |
|---|---|---|
| Channels | 128 | 256 |
| Depth | 2 | 3 |
| Channels multiple | 1,2,2,2 | 1,2,2,2 |
| Heads | 4 | 4 |
| Heads Channels | 64 | 64 |
| Attention resolution | 16 | 16 |
| Dropout | 0.1 | 0.1 |

We use the U-Net architecture from Dhariwal & Nichol (2021) for $H_\theta$ and the U-Net from Ronneberger et al. (2015) for $\gamma_\phi$. For tensor-valued time, we set the existing time embedding part of the U-Net to zero values. Details of the configurations for $H_\theta$ are provided in Table 9. For $\gamma_\phi$, we use channel configurations of [256, 512, 1024, 2048]. For $D_\psi$, we utilize four convolutional layers with 1024 channels, followed by batch normalization and leaky ReLU activation, with a sigmoid activation in the last layer. We set $M = 10$ as the default value.

## E.2 TRAINING CONFIGURATIONS

Table 10: Hyperparameters for training $H_\theta$ and path optimization.

| Hyperparameter | CIFAR-10 | | ImageNet-32 | |
|---|---|---|---|---|
| | Train $H_\theta$ | Path Opt. | Train $H_\theta$ | Path Opt. |
| Batch size | 128 | 16 | 512 | 15 |
| GPUs | 1 | 1 | 4 | 1 |
| Iterations | 400k | 200k | 250k | 200k |
| Peak LR | 2e-4 | 2e-4 | 2e-4 | 2e-4 |
| LR Scheduler | Poly decay | Poly decay | Poly decay | Poly decay |
| Warmup steps | 5k | 5k | 5k | 5k |
| Warmup steps for $D_{\theta_2}$ | - | 20k | - | 20k |

Our overall training setup is based on the code provided by Tong et al. (2023; 2024). Training is conducted on NVIDIA's RTX 3080Ti, RTX 4090, or RTX A6000 GPUs. Vanilla GAN loss in Equation 12 is employed for MTO. The Adam optimizer with $\beta_1 = 0.9$, $\beta_2 = 0.999$, weight decay of 0.0, and $\epsilon = 1e-8$ is used along with polynomial decay for learning rate scheduling throughout all training phases. An exponential moving average with a decay rate of 0.999 is also employed during all training phases. For path optimization, we evaluate FID every 10000 steps and report the lowest FID observed. Detailed configurations are provided in Table 10.

E.3 EXPERIMENTS FOR PRE-TRAINING STAGE

Table 11: Comparison of FIDs ↓ between unidimensional coefficient and multidimensional coefficient (non-LPF and LPF) for unoptimized paths using the Euler solver.

| Method \ NFE | CIFAR-10 | | | | ImageNet-32 | | | |
|---|---|---|---|---|---|---|---|---|
| | 10 | 100 | 150 | 200 | 10 | 100 | 150 | 200 |
| $\text{SI}_{\text{unidimensional}}$ | **14.43** | 4.75 | 4.51 | 4.30 | 17.72 | 8.08 | 7.79 | 7.63 |
| $\text{SI}_{\text{non-LPF}_{s=0.005}}$ | 14.59 | 3.98 | 3.74 | **3.63** | **17.41** | **6.33** | **6.21** | **6.20** |
| $\text{SI}_{\text{LPF}_{s=0.1}}$ | 15.44 | **3.77** | **3.68** | 3.75 | 17.86 | 6.63 | 6.47 | 6.44 |
| $\text{FM}_{\text{unidimensional}}$ | **13.70** | 4.52 | 4.23 | 4.07 | 16.92 | 7.78 | 7.53 | 7.38 |
| $\text{FM}_{\text{non-LPF}_{0.005}}$ | 13.81 | **3.59** | **3.42** | **3.42** | **16.85** | **6.18** | **6.03** | **6.01** |
| $\text{FM}_{\text{LPF}_{0.1}}$ | 15.13 | 3.64 | 3.57 | 3.64 | 17.52 | 6.40 | 6.27 | 6.31 |
| $\text{DDPM}_{\text{unidimensional}}$ | 98.47 | 6.64 | 4.84 | 4.10 | **111.54** | 8.13 | 7.40 | 7.14 |
| $\text{DDPM}_{\text{non-LPF}_{0.005}}$ | 74.44 | **3.77** | 5.96 | 7.84 | 139.69 | 7.67 | 12.37 | 11.70 |
| $\text{DDPM}_{\text{LPF}_{0.005}}$ | 72.23 | 4.73 | **4.11** | **3.83** | 135.48 | 6.84 | **6.51** | **6.42** |
| $\text{DDPM}_{\text{LPF}_{0.1}}$ | **71.80** | 4.46 | 6.32 | 12.60 | 142.99 | **6.70** | 8.69 | 10.91 |

Table 12: FIDs for different $\sigma$ for the Gaussian kernel in low-pass filter in $\text{SI}_{\text{LPF}_{0.1}}$ using an unoptimized path on CIFAR-10.

| $\sigma$ \ NFE | 10 | 20 | 30 | 40 | 50 | 100 | 150 | 200 |
|---|---|---|---|---|---|---|---|---|
| 0.1 | **14.89** | **8.05** | 6.49 | 5.45 | 6.53 | 9.59 | 10.67 | 11.20 |
| 1.0 | 14.92 | 8.47 | **5.32** | **4.45** | **4.68** | 6.06 | 7.01 | 7.50 |
| 2.0 | 16.25 | 9.56 | 7.56 | 6.06 | 4.72 | **3.77** | 3.95 | 4.17 |
| 4.0 | 15.44 | 9.13 | 7.39 | 6.36 | 5.59 | 3.77 | **3.68** | **3.75** |

$X_{\text{LPF}}$ denotes the hypothesis space $\gamma_\phi$ with low-pass filtering applied, while $X_{\text{non-LPF}}$ represents the hypothesis space without low-pass filtering. The parameter $s$ indicates the scale value used in these configurations. By the experiments in Table 11 and Table 12, we can identify the appropriate choice of hypothesis space that **not only maintains but also upgrades performance for pre-training.**

E.4 EXPERIMENTS FOR ADVERSARIAL TRAINING STAGE

Table 13: FIDs for path optimizations with 10 NFE Euler solver on CIFAR-10.

| Method \ $M$ | 5 | 10 | 15 | 20 | 25 | 30 |
|---|---|---|---|---|---|---|
| $\text{SI}_{\text{LPF}_{0.1}}$ | 6.89 | **4.14** | 4.42 | 5.32 | 6.11 | 5.74 |
| $\text{FM}_{\text{LPF}_{0.1}}$ | **5.93** | 6.13 | 6.70 | 6.18 | 5.97 | 6.42 |
| $\text{DDPM}_{\text{LPF}_{0.1}}$ | 10.15 | 10.04 | 9.60 | 9.04 | **8.94** | 9.19 |

Table 14: FIDs for path optimizations using $\text{SI}_{\text{LPF}_{0.1}}$ with different NFE on CIFAR-10.

| Method \ NFE | 4 | 6 | 8 | 10 |
|---|---|---|---|---|
| $\text{SI}_{\text{LPF}_{0.1}}$ | 20.59 | 6.62 | 4.85 | **4.14** |
| $\text{FM}_{\text{LPF}_{0.1}}$ | 16.42 | 8.17 | 6.56 | **6.13** |
| $\text{DDPM}_{\text{LPF}_{0.1}}$ | 72.64 | 20.13 | 13.72 | **10.04** |

Table 15: FIDs for path optimizations with 10 NFE and different inputs to $\gamma_\phi$ on CIFAR-10.

| Method \ Input | 1 | $z$ | $x_T$ |
|---|---|---|---|
| $\text{SI}_{\text{LPF}_{0.1}}$ | 7.84 | 6.48 | **4.14** |
| $\text{FM}_{\text{LPF}_{0.1}}$ | 9.20 | 9.06 | **6.13** |
| $\text{DDPM}_{\text{LPF}_{0.1}}$ | 26.09 | 23.31 | **10.04** |

Table 16: FIDs for path optimizations with 10 NFE and SI trained using various hypothesis space $\gamma_\phi$ on CIFAR-10.

| Method \ $M$ | 5 | 10 | 15 | 20 |
|---|---|---|---|---|
| $t$ | 10.20 | 9.75 | 11.30 | 11.53 |
| non-LPF$_{0.005}$ | 6.60 | 4.79 | 4.45 | 5.28 |
| LPF$_{0.005}$ | 7.37 | 4.42 | 4.26 | 5.31 |
| LPF$_{0.1}$ | 7.21 | **4.14** | 5.59 | 5.32 |

To validate that the extra $\phi$ parameterization and optimization have practical benefits, **MTO in these results is obtained by training only $\phi$ while keeping $\theta$ frozen**, supporting our novelty and contribution in parameterizing the adaptive multidimensional coefficient and performing MTO. We achieve 4.14 and 7.06 FID values in CIFAR-10 and ImageNet-32, respectively, with 10 NFEs using SI$_{\text{LPF}_{0.1}}$. As shown in Table 14, MTO can be applied to different sampling configurations. In Table 15, we validate the use of the adaptive multidimensional coefficient conditioned on the starting point of the differential equation, $x_T$. Using $x_T$ as the input for $\gamma_\phi$ consistently achieves better performance across three different methodologies, providing empirical evidence for the advantage of the adaptive multidimensional coefficient over using the same coefficient for all different $x_T$. In Table 16, we can identify the appropriate choice for the hypothesis space for MTO.

## F    GENERATED SAMPLES.

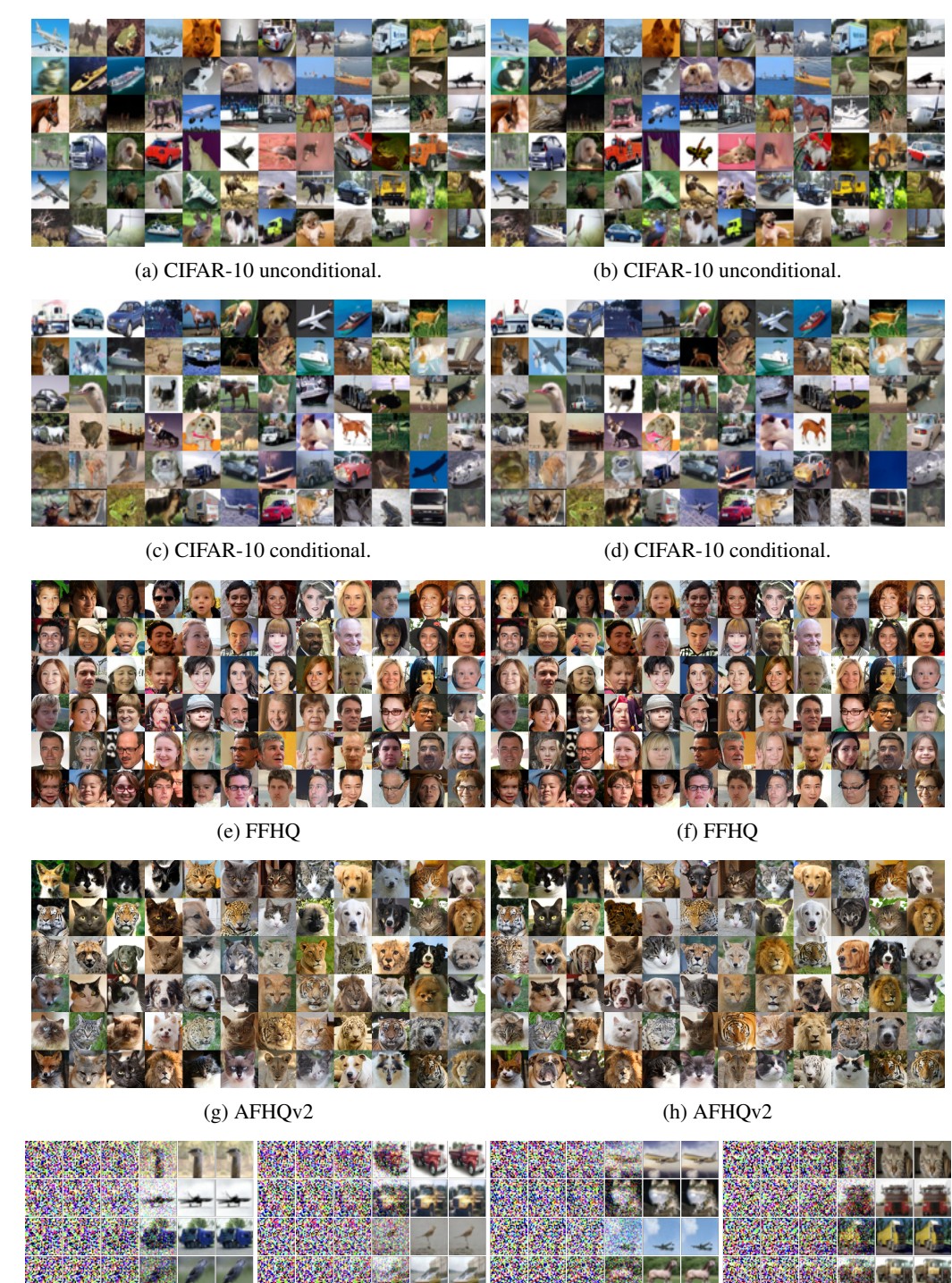

(a) CIFAR-10 unconditional.
(b) CIFAR-10 unconditional.

(c) CIFAR-10 conditional.
(d) CIFAR-10 conditional.

(e) FFHQ
(f) FFHQ

(g) AFHQv2
(h) AFHQv2

(i) Optimized trajectories for CIFAR-10 conditional generation.

Figure 11: EDM (left) and EDM—MTO's (right) generated samples on various datasets.

