# OpenReview forum: "Multidimensional Trajectory Optimization for Flow and Diffusion"
_ICLR.cc/2025/Conference — Submitted to ICLR 2025_

### Official Review · Reviewer_dLdR · 2024-11-01

**Soundness:** 3
**Presentation:** 2
**Contribution:** 2
**Rating:** 5
**Confidence:** 3

**Summary:**

This paper proposes an extension of flow/diffusion models which allows for the interpolant coefficient to now be multi-dimensional, i.e. with one coefficient per data dimension. The authors propose to optimize these coefficients to obtain learnable trajectories in order to improve model performance. To do so, a specific parametric form for the trajectories is specified, followed by adversarial training. The proposed method is evaluated on a few low-dimensional image datasets.

**Strengths:**

- The idea of having learnable trajectories is interesting
- The experiments show that the learnable trajectories can indeed give you better performance

**Weaknesses:**

- The paper is a little hard to read, and could generally benefit from additional exposition and improved clarity.
     - For example, in definition 1, what is the reason behind the boundary conditions on $\gamma$? This might generally be clear to readers well-versed in flows and diffusion, but explaining this would help the clarity in the paper.
     - Another place where clarity is lacking is that in Def. 1 the coefficients $\gamma$ are not allowed to depend on $x_1$, but only a few lines later the authors now allow $\gamma$ to depend on $x_1$.
     - Or, in lines 125, the authors use $\rho_1$ as the noise distribution for EDM, but then in line $130$ are using $\rho_T$ to denote the noise distribution.
     - I found the description of the adversarial training in Section 4.4 to be generally unclear, e.g. a discriminator $D_\psi$ is introduced in the equations but never really explained.
- One of the major benefits of flow and diffusion models is that they are simulation-free during training, making training generally straightforward to carry out. This paper moves in the opposite direction and now requires not only simulation from the model, but also a fairly complex adversarial setup. I would be interested in seeing an evaluation of the training cost of the proposed method, which I'd guess is significantly higher than other flow/diffusion methods.
- The datasets considered are all quite low-dimensional (2d synthetic datasets, CIFAR-10, and some 64x64 image datasets). This makes me wonder if the proposed method is scalable to higher-dimensional datasets.

**Questions:**

- The proposed method generally performs worse than methods based on diffusion + distillation in Tables 3 and 4, despite having a 5x higher NFE than these baselines. Do the authors have an explanation for why this is the case?
- There is some existing work on learning trajectories in diffusion models. e.g. [Where to Diffuse...](https://arxiv.org/abs/2302.07261) and [Neural Flow Diffusion Models](https://arxiv.org/abs/2404.12940v2). What is the relationship between these methods (and other existing learnable diffusions) and this work?

---

> ### Author Response · Authors · 2024-11-17
>
> First of all, we really appreciate your insights and valuable feedback. We are especially grateful for recognizing the following strengths of our work, which we summarize below:
>
> [$\textbf{STRENGTHS}$]
>
> We thank Reviewer dLdR for encouraging feedback. Reviewer dLdR highlighted the novelty of having learnable trajectories, emphasizing how this idea introduces an interesting dimension to trajectory optimization. We are pleased that Reviewer dLdR recognized the value of our approach in improving generative model performance through learnable trajectories. Additionally, Reviewer dLdR appreciated our experimental results, which demonstrate that learnable trajectories can indeed lead to better performance. This acknowledgment of our experiments’ effectiveness reinforces the practical utility of our proposed methodology. We are glad that Reviewer dLdR found our contributions both innovative and impactful.
>
> [$\textbf{WEAKNESSES}$]
>
> [$\textbf{Weakness 1.}$] The paper is a little hard to read, and could generally benefit from additional exposition and improved clarity
>
> [$\textbf{1. Paper clarity issues}$]
>
> We totally agree with your comments. Sorry for the poor explanation. We revised all parts you have mentioned as below.
>
> First, to clarify the boundary condition for the multidimensional coefficient, we added the sentence:
>
> "Boundary conditions written above ensure that $x(t)$ becomes $x_0$ and $x_T$ for $t=0$ and $t=T$, which is a requirement for transportation. Values $k$ and $T$ for boundary conditions vary based on the task."
>
> Second, for rigorous definition and better readability, we introduced a new definition called the adaptive multidimensional coefficient, which captures the concept of adaptability for various inference trajectories. The definition is written below:
>
> $\textbf{Definition 2: Adaptive Multidimensional Coefficient}$
> For $t \in [0, T]$ and inference trajectory $x_{\theta, \phi}(t)$, the adaptive multidimensional coefficient $\gamma_\phi(t, x_{\theta, \phi}(t)):[0, T] \times \mathbb{R}^d \rightarrow \mathbb{R}^{2 \times d}$ is parameterized by $\phi$. Boundary conditions follow Definition 1 (Multidimensional Coefficient), with $\gamma_\phi \in C^1([0, T], \mathbb{R}^{2 \times d})$.
>
> This definition describes the trajectory-conditioned multidimensional coefficient. To reduce computational costs in calculating $\gamma_\phi$, we use only $t = T$ for $x_{\theta, \phi}(t)$ in $\gamma_\phi(t, x_{\theta, \phi}(t))$ rather than the inference trajectory at multiple time points. This approach allows us to compute $\gamma_\phi$ across the entire inference time schedule $\tau = \{ t_0, \ldots, t_N \}$ with a single function evaluation before initiating transportation. By using the adaptive multidimensional coefficient, we can address multidimensional trajectory optimization more rigorously.
>
> Third, we unified all the notations for noise distribution and samples to $x_T \sim \rho_T$.
>
> Lastly, we revised the adversarial training section with more details based on your comments:
>
> "
> Following GAN [1], we use an adversarial loss to minimize Equation 6. Specifically, we employ the StyleGAN-XL [2] discriminator for $D_\psi$ with hinge loss [3] for $\theta$, $\phi$, and $\psi$ (discriminator parameters), as used in [4]. In our method, $\theta$ and $\phi$ in $G_{\theta, \phi}$ aim to deceive $D_\psi$. The loss function for $\theta$ is:
>
> $L_{\theta} = - E_{x_1 \sim \rho_1}[D_\psi \left( H_\theta(t, x(t), \gamma_\phi(t, z)) \right) ], \quad x(t) = x_0 + \gamma_{1, \phi}(t,z) \odot x_1, \quad z \sim \rho_T,$
>
> where $L_\theta$ indicates that the gradient is only calculated with respect to $\theta$. Since $\theta$ only needs to handle elements in $\{\gamma_\phi(t, x_T) \mid t \in [0, T], x_T \sim \rho_T\}$ and not other elements in $\Gamma_h$, $\theta$ is trained exclusively on $\gamma_\phi$, helping reduce the load on $H_\theta$. The loss functions for $\phi$ and $\psi$ are:
>
> $L_\phi = - E_{x_1 \sim \rho_1}[D_\psi \left( G(\tau, x_1,  v_{\theta, \phi}) \right) ],$
>
> $L_\psi = E_{x_0 \sim \rho_0}[\max(0, 1 - D_\psi(x_0))] + E_{x_1 \sim \rho_1}[\max(0, 1 + D_\psi(G(\tau, x_1,  v_{\theta, \phi})))].$
>
> Here, $L_\phi$ and $L_\psi$ indicate that gradients are calculated with respect to $\phi$ and $\psi$, respectively, as shown in Figure 3. With these loss functions, $\phi$ is optimized to find better trajectories, while $\theta$ adapts to $\gamma_\phi$, which is sparser than $\Gamma_h$.
> "
>
> With these clarifications, we hope that the paper's readability is significantly improved. Thank you for pointing out these specific issues.

---

> ### Author Response · Authors · 2024-11-17
>
> [$\textbf{Weakness 2.}$] One of the major benefits of flow and diffusion models is that they are simulation-free during training, making training generally straightforward to carry out. This paper moves in the opposite direction and now requires not only simulation from the model, but also a fairly complex adversarial setup. I would be interested in seeing an evaluation of the training cost of the proposed method, which I'd guess is significantly higher than other flow/diffusion methods.
>
> [$\textbf{Weakness 3.}$] The datasets considered are all quite low-dimensional (2d synthetic datasets, CIFAR-10, and some 64x64 image datasets). This makes me wonder if the proposed method is scalable to higher-dimensional datasets.
>
> [$\textbf{2 and 3. Illustrating the training cost of simulation training to highlight our efficiency and scalability}$]
>
> After reading your comments, we realized that two points were insufficiently discussed:
> 1. Why we adopt the combination of simulation-free and simulation-based end-to-end methods.
> 2. Training cost, efficiency, and scalability issues.
>
> We addressed these topics specifically as follows:
>
> $\textbf{Reasons for combining simulation-free and simulation-based end-to-end methods}$
>
> We tried to answer the main question:
>
> “Given a differential equation solver with fixed configurations, which multidimensional trajectory yields optimal performance in terms of final transportation quality for a given starting point of the differential equation?”
>
> This question highlights a trade-off inherent in diffusion models, where simulation-free objectives—while beneficial in reducing training costs—limit adaptability in trajectory optimization concerning output quality. This flexibility is retained in simulation-dynamics [5]. To enable trajectory optimization while maintaining simulation-free objectives for training cost efficiency, prior approaches have relied on pre-defined trajectory properties, such as straightness [6, 7], to minimize numerical error. However, such pre-defined properties for the trajectories diverge from true transportation optimality, as they fail to account for the sole measure of optimality in our perspective: the final quality of transportation.
>
> We reintroduce trajectory adaptability by employing simulation dynamics combined with adversarial training [5], defining trajectory optimality solely based on the final generative output under fixed solver configurations. To effectively leverage the advantages of simulation-based objectives—which provide adaptability (in terms of inference trajectories) and flexibility (in terms of multidimensionality) of trajectories—while mitigating their inefficiencies in training, we use simulation-based objectives only for $\phi$ after pre-training $\theta$ with a simulation-free objective. This approach makes trajectory optimization feasible in terms of efficiency and scalability. Our experiments demonstrate that trajectories optimized through this approach significantly improve the performance of flow and diffusion models.
>
> $\textbf{Training efficiency and scalability issues}$
>
> We found that the training efficiency of our methodology is significantly better than other distillation methods, such as GDD [8]. We evaluated the number of images required for training and the training time, comparing these with GDD and GDD-I [8]. Remarkably, our approach for MTO requires far fewer images for training across all datasets compared to GDD-I, a method known for its efficiency in diffusion distillation:
>
> CIFAR-10: 5000kimg (GDD-I) vs. 1382kimg (Ours)
>
> FFHQ: 10000kimg (GDD-I) vs. 106kimg (Ours)
>
> AFHQv2: 10000kimg (GDD-I) vs. 955kimg (Ours)
>
> The training times for our method are 10, 2, and 6 hours for CIFAR-10, FFHQ, and AFHQv2, respectively, which are also comparatively low. Additionally, FID significantly decreases in the early stages of training, as illustrated in Figure 6 (we modified the x-axis from iteration number to kimg for improved readability). The primary computational cost arises from VRAM requirements; however, training remains practically feasible. Good performance is achieved with just 5 NFE, which is relatively low. All experiments for adversarial training were conducted on GPUs with 48GB of VRAM—less than the 80GB VRAM GPUs often used in related works.
>
> These results underscore the effectiveness of simulation-based end-to-end optimality while highlighting the strength of combining simulation-free and simulation-based methodologies, leveraging the advantages of each while mitigating their limitations. Considering these aspects, we estimate that our adversarial approach for MTO is scalable to larger datasets while maintaining efficiency.

---

> ### Author Response · Authors · 2024-11-17
>
> Regarding experiments on large and high-dimensional datasets like ImageNet, we completely agree with your point. We tried our best to validate our methodology's scalability and effectiveness but couldn't afford to train the EDM-based $H_\theta$ from scratch for ImageNet-64, which requires 32 A100 GPUs and several weeks of training for a fair comparison. We sincerely aim to experiment with our methods on large datasets like ImageNet and video datasets in future work. Thank you again for pointing out this important perspective.
>
> [$\textbf{Question 1.}$] The proposed method generally performs worse than methods based on diffusion + distillation in Tables 3 and 4, despite having a 5x higher NFE than these baselines. Do the authors have an explanation for why this is the case?
>
> [$\textbf{1. Reasons for the results in FFHQ and AFHQv2}$]
>
> We apologize for not thoroughly discussing the results for FFHQ and AFHQv2. Thank you for pointing out this important aspect.
>
> First, we want to emphasize that comparing the adversarial approach to MTO with diffusion distillation is not entirely fair, as these methodologies are fundamentally different and can even be complementary. However, due to the limited availability of comparable experiments from diffusion distillation and trajectory optimization on FFHQ and AFHQv2, it was necessary to compare our results with these methods.
>
> Second, we aim to show the relationship between data dimensionality, model size for $\gamma_\phi$, and NFE for performance. By using the same model size for $\gamma_\phi$ (specific configurations are written in Appendix C.2) and the same NFE for all datasets (note that baseline EDM uses more than twice the NFE for FFHQ and AFHQv2 compared to CIFAR-10), we found that FFHQ and AFHQv2’s higher dimensionality (64$\times$64 resolution) compared to CIFAR-10 (32$\times$32 resolution) imposes additional burden on $\gamma_\phi$ to find adaptive trajectories, requiring more NFE to achieve comparable FID values.
>
> These observations indicate that achieving significant performance improvements in FFHQ and AFHQv2 similar to CIFAR-10 would require either a larger model for $\gamma_\phi$ or more NFE during training. As noted in the $\textbf{Training efficiency and scalability issues}$ section, early convergence observed in FFHQ and AFHQv2 signals that a larger $\gamma_\phi$ model could improve performance further.
>
> Future works could resolve these training efficiency and performance challenges in two ways:
> 1. Designing a more efficient $\gamma_\phi$ architecture to reduce computational costs (e.g., model size and NFE).
> 2. Developing more efficient training strategies to enhance performance.
>
> Based on this, we added senteneces to the limitation section as below:
>
> "Additionally, improving the design of $\gamma_\phi$ could enhance efficiency. As shown in Table 3 and Table 4, our method’s FID is higher than distillation methods for larger datasets. This may stem from using the same model size and NFE settings as in smaller datasets like CIFAR-10. Refining $\gamma_\phi$ could reduce model size and NFE while maintaining performance."
>
> Thank you for pointing out these important points.
>
> [$\textbf{Question 2.}$] There is some existing work on learning trajectories in diffusion models. e.g. Where to Diffuse... and Neural Flow Diffusion Models. What is the relationship between these methods (and other existing learnable diffusions) and this work?
>
> [$\textbf{2. Comparing with existing trajectory optimization works}$]
>
> We sincerely appreciate your effort in identifying relevant related works. We compare MTO and our adversarial approach for solving MTO with other research in terms of four key aspects:
>
> (i) {Does $\phi$ exhibit multidimensionality? (multidimensional coefficient)
>
> (ii) Is $\phi$ trajectory-dependent? (adaptive multidimensional coefficient as defined above)
>
> (iii) Are $\theta$ and $\phi$ jointly learnable?
>
> (iv) Is optimality defined by the final transportation quality?
>
> WTH (Where to diffuse...) [10] defines optimality through a fixed sequence of diffusion steps aimed at reducing inference complexity rather than focusing on the quality of final samples. Additionally, NFDM (Neural Flow Diffusion Models) [11] introduces neural flow models that implicitly set trajectory optimality within the diffusion process, aiming to generate high-quality samples without explicit trajectory adjustment post-training.
>
> While these methods satisfy (iii) and provide diverse perspectives on trajectory optimality, there are two significant differences between these methods and ours:
>
> 1. We calculate trajectory optimality solely based on the final transportation quality by end-to-end training (iv), which is crucial in generative modeling.
>
> 2. None of these methods achieve full trajectory flexibility regarding multidimensionality (i) and adaptability (ii) for different inference trajectories.

---

> ### Author Response · Authors · 2024-11-17
>
> We revised the trajectory optimization section in the related works to reflect these distinctions more clearly. The revised content is as follows:
>
> "
> $\textbf{Trajectory Optimizations in Flow and Diffusion}$
> Various trajectory optimization approaches pre-define optimality without relying on final transportation quality. Approaches such as [5, 6] define straightness as the optimality criterion and optimize trajectories by maintaining the consistency of $(x_0, x_1)$ for training flow and diffusion models, aligning with an optimal transport perspective. Another example is [10], which defines optimality through a fixed sequence of diffusion steps aimed at reducing inference complexity rather than focusing on the quality of final samples. Additionally, [11] introduces neural flow models that implicitly set trajectory optimality within the diffusion process, aiming to generate high-quality samples without explicit trajectory adjustment post-training. There are also approaches that refine trajectories after training, such as [12], where optimality is defined by minimizing the trajectory length in the Wasserstein-2 metric, focusing on a shortest-distance criterion. Despite these diverse perspectives on trajectory optimality, there are two significant differences between these methods and ours. First, we calculate the optimality of the trajectory solely based on the final transportation quality, which is a crucial factor in generative modeling. Second, none of these methods achieve full flexibility of the trajectory on two fronts: multidimensionality and adaptability with respect to different inference trajectories.
> "
>
> This revision ensures that the differences between our approach and existing methods are clearly articulated and the uniqueness of our contributions is highlighted. Thank you for your insightful comments that helped us clarify these points.
>
> We once again sincerely thank to you for your thoughtful feedback and valuable insights. Your suggestions have greatly helped us to clarify and improve the presentation, scalability, and technical rigor of our work. We have revised the paper thoroughly to address all the comments and have uploaded a new version that reflects these changes.
>
> If you have any further questions or comments, we would be happy to address them without hesitation. Thank you again for your constructive and encouraging feedback.
>
> [1] Ian Goodfellow, Jean Pouget-Abadie, Mehdi Mirza, Bing Xu, David Warde-Farley, Sher- jil Ozair, Aaron Courville, and Yoshua Bengio. Generative adversarial nets, 2014.
>
> [2] Axel Sauer, Katja Schwarz, and Andreas Geiger. Stylegan-xl: Scaling stylegan to large diverse datasets, 2022.
>
> [3] Jae Hyun Lim and Jong Chul Ye. Geometric gan, 2017.
>
> [4] Dongjun Kim, Chieh-Hsin Lai, Wei-Hsiang Liao, Naoki Murata, Yuhta Takida, Toshimitsu Uesaka, Yutong He, Yuki Mitsufuji, and Stefano Ermon. Consistency trajectory models: Learning probability flow ODE trajectory of diffusion, 2024.
>
> [5] Ricky T. Q. Chen, Yulia Rubanova, Jesse Bettencourt, and David K Duvenaud. Neural ordinary differential equations, 2018.
>
> [6] Alexander Tong, Kilian FATRAS, Nikolay Malkin, Guillaume Huguet, Yanlei Zhang, Jarrid Rector Brooks, Guy Wolf, and Yoshua Bengio. Improving and generalizing flow-based generative models with minibatch optimal transport, 2024.
>
> [7] Xingchao Liu, Chengyue Gong, and qiang liu. Flow straight and fast: Learning to generate and transfer data with rectified flow, 2023.
>
> [8] Bowen Zheng and Tianming Yang. Diffusion models are innate one-step generators, 2024.
>
> [9] Ian Goodfellow, Jean Pouget-Abadie, Mehdi Mirza, Bing Xu, David Warde-Farley, Sher- jil Ozair, Aaron Courville, and Yoshua Bengio. Generative adversarial nets, 2014.
>
> [10] Raghav Singhal, Mark Goldstein, and Rajesh Ranganath. Where to diffuse, how to diffuse, and how to get back: Automated learning for multivariate diffusions, 2023.
>
> [11] Grigory Bartosh, Dmitry Vetrov, and Christian A. Naesseth. Neural flow diffusion models: Learnable forward process for improved diffusion modelling, 2024.
>
> [12] Michael Samuel Albergo, Nicholas Matthew Boffi, Michael Lindsey, and Eric Vanden-Eijnden. Multimarginal generative modeling with stochastic interpolants, 2024

---

> > ### Comment · Reviewer_dLdR · 2024-11-27
> >
> > I thank the authors for their detailed and constructive response to my review. The new experiments and discussion relating to the efficiency and scalability are appreciated, and adequately address my concerns regarding these points.
> >
> > Overall, I think the approach proposed in the paper is interesting. However, I still find the updated version of the paper quite difficult to read and lacking in clarity, and for this reason I'd like to maintain my score. I think this could be a strong paper if the authors re-structured their paper to increase the clarity. I do not think that minor changes in the submission will be sufficient to make the approach clear.
> >
> > As mentioned in my original review, one place where this is particularly relevant is in the role of the adversarial training. For instance, the authors write things like:
> > - "Our approach employs simulation dynamics and adversarial training to optimize these inference trajectories."
> > - "We reintroduce trajectory adaptability by employing simulation dynamics combined with adversarial
> > training (Goodfellow et al., 2014), defining trajectory optimality based solely on the final generative output..."
> >
> > but do not explain clearly what exactly the adversarial training is meant to achieve (i.e., training a model which can be used for many different coefficients?).
> >
> > Furthermore the authors write that the optimal $\theta, \phi$ as posed in Equation 5 are found via a minimization problem -- which does not require adversarial optimization. The section on adversarial training (page 6) does not explicitly state what the min/max problem is.
> >
> > I believe (but am not confident in my understanding) that the authors are doing something like pre-training $\theta$ by solving a problem of the form $\min_\theta \max_\phi$.... it would be very helpful to explicitly write this out though, rather than jumping immediately into expressions for the Euler solver, etc.
> >
> > Generally, I feel like too much emphasis in the writing is placed on technical, implementation-level details that would be better suited for an appendix -- thus leaving too little space for exposition and explaining the key ideas of the approach more clearly. - - - As an example, Equations 9 and 10 could likely be placed in an appendix.
> > - Similarly e.g. lines 179-200 could be significantly condensed -- I don't think there is any need to go into such level of detail regarding minimizing the loss on a finite set of samples (which is the approach taken by ~all machine learning work...)
> > - Again, in Equation 12, I am not sure so much space (and dense symbols) are needed just to explain the Euler solver, which is a standard/well-known thing in the diffusion literature

---

> ### Author Response · Authors · 2024-11-28
> **Thank Reviewer dLdR for the responses**
>
> First of all, we truly appreciate your concise feedback for clarifying and improving the writing of our work.
>
> 1. $\textbf{Explain our methodology precisely}$
>
> We thank Reviewer dLdR for pointing out unclear and confusing parts in our paper that might have caused misunderstandings. Here, we aim to provide a precise explanation of our methodology.
>
> The training is divided into two processes, with adversarial training applied only during the second process:
>
> - $\textbf{1st process (Pre-training stage, non-adversarial)}$: We train $\theta$ using various randomly sampled coefficients from the hypothesis space $\Gamma_h$, which we specifically designed to exclude crude coefficients. This process employs the same loss function and training methodology used in standard diffusion models. After this pre-training stage, $H_\theta$ can handle denoising tasks with $\gamma \sim \Gamma_h$, preparing it for MTO.
>
> - $\textbf{2nd process (MTO stage, adversarial)}$: During this stage, we train $G_{\theta, \phi}$ as a generator, composed of $H_\theta$ and $\gamma_\phi$, using a discriminator $D_\psi$. The adversarial training is conducted between {$\theta, \phi$} <-> {$\psi$}.
>
> Based on this understanding, we clarified the points you mentioned as follows:
>
> 2. $\textbf{Clarifying the methodology, especially for adversarial training}$
>
> We appreciate your feedback on the lack of clarity regarding the adversarial training section. To address this:
>
> - We revised sentences like "Our approach employs simulation dynamics and adversarial training to optimize these inference trajectories" to "Our approach pre-trains flow and diffusion models with various coefficients sampled from a hypothesis space and subsequently optimizes inference trajectories through adversarial training of a generator comprising the flow or diffusion model and the parameterized coefficient" (lines 17–20, 212-214, 516–518).
>
> - We refined the explanation: "We reintroduce trajectory adaptability by employing simulation dynamics combined with adversarial training (Goodfellow et al., 2014), defining trajectory optimality based solely on the final generative output..." to "We reintroduce trajectory adaptability by employing simulation dynamics combined with adversarial training, defining trajectory optimality based solely on the final generative output under fixed solver configurations. Specifically, first, we pre-train a diffusion model $H_\theta$ with randomly sampled multidimensional coefficients $\gamma$ from a well-designed hypothesis space. This pre-training enables the flow or diffusion model to handle various coefficients, preparing it for the trajectory optimization stage. Next, we introduce the parameterized adaptive multidimensional coefficient $\gamma_\phi$ to compose a flow or diffusion-based generator $G_{\theta, \phi}$, which produces $x_{\text{est}, \theta, \phi}$ through simulation dynamics. A discriminator $D_\psi$ evaluates the generated samples $x_{\text{est}, \theta, \phi}$ to optimize both $\theta$ and $\phi$, a process we term multidimensional trajectory optimization" (lines 54–62).
>
> - To avoid confusion about the two-stage training process, we explicitly added the pre-training loss term in Section 4.3 (lines 284–289).
>
> - We included a min-max equation from GANs (Equation (12)) to clarify the connection between MTO and the adversarial approach to MTO.
>
> 3. $\textbf{Move technical details to Appendix}$
>
> - We agree that technical, implementation-level details should generally be moved to the appendix to focus on the main ideas in the paper. However, we believe Equations 9 and 10 should remain in the main text, as they are crucial for explaining our hypothesis space design. This decision is particularly relevant in addressing Reviewer xR7Z’s question: "In the pretraining stage, how exactly are randomly sampled trajectories obtained? Do you still use the proposed parameterized family of coefficients in this stage? If so, which parameters are set to random, and what distributions do they follow?" These details are critical for understanding the performance and stability benefits provided by excluding crude coefficients in our hypothesis space design. That said, we agreed to minimize technical explanations in the main text. The choice of $b_m(t)$ is now briefly explained in line 256, with detailed discussions moved to Appendix A. Other technical sections, such as "lines 179–200 (finite set of samples)" and the explanation of the Euler solver, have been condensed and moved to Appendix A.

---

> ### Author Response · Authors · 2024-11-28
> **Thank Reviewer dLdR for the responses**
>
> 4. $\textbf{Revision for our comment}$
>
> Finally, we revisited our response to your initial review question: "In line 125, the authors use $\rho_1$ as the noise distribution for EDM, but in line 130, $\rho_T$ is used instead." Using $x_1 \sim \rho_1 = \mathcal{N}(0, I)$ and $\rho_T = \mathcal{N}(0, T^2I)$ is correct, as the former applies to the pre-training stage and the latter to the inference stage. We sample $x_0, x_1 \sim \rho_0, \rho_1$ to train the flow or diffusion model and sample $x_T \sim \rho_T$ for transportation. We apologize for the repeated revisions.
>
>
> Overall, we thank you again for your beneficial feedback, which has helped us clarify our writing.
>
> 5. $\textbf{Additional experiments}$
>
> We also conducted additional experiments to validate the empirical benefits of our methodology, which are now organized in Appendix E to further support the contributions of our work. We want to highlight three key points:
>
> 1. Pre-training with the multidimensional coefficient not only preserves but also enhances the performance of flow and diffusion models.
> 2. All MTO experiments in Appendix E were conducted by training only $\phi$, while keeping $\theta$ frozen.
> 3. Even with a small 4-layer discriminator network and Vanilla GAN Loss, we achieved comparable FID performance on SI.
>
> Thank you for your attention to these updates.

---

> > ### Author Response · Authors · 2024-12-03
> > **Summary of Revisions and Final Check**
> >
> > We have summarized our revisions and key points in a top comment for all reviewers. If there are any additional clarifications or points you’d like us to address before the discussion closes, please let us know. Thank you again for your thoughtful feedback.

---

### Official Review · Reviewer_xR7Z · 2024-11-04

**Soundness:** 3
**Presentation:** 3
**Contribution:** 3
**Rating:** 6
**Confidence:** 4

**Summary:**

This paper proposes a novel multi-dimensional parameterization of diffusion/flow coefficients along with an adversarial learning objective to optimize the sample trajectories. Experiments on a synthetic dataset and several real-world image datasets are conducted to verify the effectiveness of the proposed method.

**Strengths:**

- The idea of learning multi-dimensional diffusion/flow coefficients is novel. The proposed parameterized family of the coefficients not only ensures the boundary condition but also permits efficient learning, as suggested by the empirical results. I could imagine follow-up work that further explores the design space.

&nbsp;

- The experimental results look promising. The proposed method consistently achieves lower Wasserstein distances in the synthetic case and is on par with or slightly better on the challenging real-world image generation tasks.

&nbsp;

- The paper is clearly written, except that some technical details are sparse (possibly due to the space limit).

**Weaknesses:**

- For a better understanding of the design of the adversarial loss, I’d like to hear authors’ feedback on the following questions. Moreover, it would be great if the authors could improve the writing in Section 4.4.

  1) The loss $L_{\psi}$ in Eq. (14) is also a function of $\phi$ and $\theta$ since it takes generated samples as input. Does the notation $L_{\psi}$ mean that the gradient is stopped so that this loss is only with respect to $\psi$? Similar questions apply to $L_{\phi}$ and $L_{\theta}$. This part is not explained and can not be deduced from Algorithm 1 since Line 12 of Algorithm 1 is also vague.

  2) Minimizing the loss $L_{\psi}$ with respect to $\psi$ encourages the discriminator $D_{\psi}$ outputs $\ge 1$ value for $x_0 \sim \rho_0$ and $\le -1$ value for $x_1 \sim \rho_1$. Minimizing the loss $L_{\phi}$ with respect to $\phi$ encourages the generator to make the discriminator output values for $x_1 \sim \rho_1$ that are as large as possible. This contradicts the effect of minimizing $L_{\psi}$, which makes sense following the adversarial principle. However, minimizing $L{\theta}$ is hard to interpret. To follow the adversarial principle, i.e., $L_{\theta}$ and $L_{\psi}$ are contradictory, one would hope the model $H_{\theta}$ makes the discriminator $D_{\psi}$ outputs $\le 1$ value for $x_0 \sim \rho_0$. But the current design actually encourages the discriminator $D_{\psi}$ output value as large as possible for $x_t$. Can you explain the rationale of the current design?

&nbsp;

- While optimizing the coefficients $\gamma$ via the generator $G$, it seems that backpropagation through the ODE solver is inevitable. This is also shown in Fig. 3. Given the high computational cost, could you comment on the scalability of your method?

&nbsp;

- In the pretraining stage, how exactly are randomly sampled trajectories obtained? Do you still use the proposed parameterized family of coefficients $\gamma$ in this stage? If so, which parameters are set to random, and what distributions do they follow? How do you ensure the pre-training does not degrade performance much and effectively explores the parameter space?

&nbsp;

- The authors do not explain the design choice of the discriminator $D_{\psi}$ in Section 4.4 when it was first introduced.

&nbsp;

- Can you discuss the motivation for using hinge loss in the adversarial training in more detail? The authors just follow the previous work in GAN training without giving motivation.

&nbsp;

- In Section 5.2, the authors mention that the NFE of the proposed method is 6 (+1 for $\gamma_{\phi}$). What exactly does this mean? Why “+1 for $\gamma_{\phi}$” rather than using learned $\gamma_{\phi}$ through the whole sampling?

&nbsp;

- In Table 2-4, when comparing with other distillation methods, it would be more fair to run other methods under the same NFE. For example, how do CD and CTM perform with 5 or 6 NFE?

&nbsp;

- It would be great to add some comments in the caption of Figure 2 to help better explain the difference and illustrate the idea.

**Questions:**

Please see my comments in the weakness section.

---

> ### Author Response · Authors · 2024-11-16
>
> First of all, we really appreciate your insights and valuable feedback. We are especially grateful for recognizing the following strengths of our work, which we summarize below:
>
> [$\textbf{STRENGTHS}$]
>
> We thank Reviewer xR7Z for insightful and encouraging comments! Reviewer xR7Z recognized the novelty of learning multidimensional diffusion/flow coefficients and appreciated the design of our parameterized family of coefficients. Reviewer xR7Z noted how our approach not only ensures boundary conditions but also enables efficient learning, as reflected in our empirical results. We are excited that Reviewer xR7Z sees potential for future follow-up work that explores the design space further. Additionally, Reviewer xR7Z found our experimental results promising. Reviewer xR7Z highlighted that our method consistently achieves lower Wasserstein distances in synthetic cases and performs on par with or slightly better than baseline methods on challenging real-world image generation tasks. We are also pleased that Reviewer xR7Z found our paper to be clearly written, despite some technical details being sparse due to space limitations.
>
> Below, we address the weaknesses mentioned in your comments in detail:
>
> [$\textbf{WEAKNESSES}$]
>
> [$\textbf{Weakness 1.}$] For a better understanding of the design of the adversarial loss, I’d like to hear authors’ feedback on the following questions. Moreover, it would be great if the authors could improve the writing in Section 4.4.
>
> $\textbf{[1-1. Ambiguity of the loss function notation]}$
>
> You are absolutely right, and we acknowledge the vagueness in the notation. We have revised this and added further explanation, such as: "where $L_\theta$ indicates that the gradient is only calculated with respect to $\theta$," and "where $L_\phi$ and $L_\psi$ indicate that gradients are calculated with respect to $\phi$ and $\psi$, respectively."
>
> $\textbf{[1-2. Clarifying the adversarial training section]}$
>
> There seems to be some misunderstanding in this part. First, we apologize for the poor explanation. In our method, not only $\phi$ but also $\theta$ in $G_{\theta, \phi}$ aims to deceive $D_\psi$. $\theta$ and $\phi$ are on the same side to fool the discriminator. To improve clarity, we have added a sentence "In our method, $\theta$ and $\phi$ in $G_{\theta, \phi}$ aim to deceive $D_\psi$." to the paper explaining this point.
>
> [$\textbf{Weakness 2.}$] While optimizing the coefficients $\gamma$ via the generator $G$, it seems that backpropagation through the ODE solver is inevitable. This is also shown in Fig. 3. Given the high computational cost, could you comment on the scalability of your method?
>
> $\textbf{[2. Assessing the efficiency and scalability of our method]}$
>
> Thanks to your insightful comments, we evaluated the number of images required for training and the training time, comparing these with GDD [1]  and GDD-I [1] . Remarkably, our approach for MTO requires far fewer images for training across all datasets compared to GDD-I, a method known for its efficiency in diffusion distillation:
>
> CIFAR-10: 5000kimg (GDD-I) vs. 1382kimg (Ours)
>
> FFHQ: 10000kimg (GDD-I) vs. 106kimg (Ours)
>
> AFHQv2: 10000kimg (GDD-I) vs. 955kimg (Ours)
>
> The training times for our method are 10, 2, and 6 hours for CIFAR-10, FFHQ, and AFHQv2, respectively, which are also comparatively low. Additionally, FID significantly decreases in the early stages of training, as illustrated in Figure 6 (We modify the x-axis from iteration number to kimg for improved readability.) The primary computational cost arises from VRAM requirements; however, training remains practically feasible. Good performance is achieved with just 5 NFE, which is relatively low. All experiments for adversarial training were conducted on GPUs with 48GB of VRAM-less than the 80GB VRAM GPUs often used in related works. These results underscore the effectiveness of simulation-based end-to-end optimality while highlighting the strength of combining simulation-free and simulation-based methodologies, leveraging the advantages of each while mitigating their limitations. Considering these aspects, we estimate that our adversarial approach for MTO is scalable to larger datasets while maintaining efficiency. We have added a new paragraph like above in the experiments section to address these scalability issues. Thank you again for your comment.

---

> ### Author Response · Authors · 2024-11-16
>
> [$\textbf{Weakness 3.}$] In the pretraining stage, how exactly are randomly sampled trajectories obtained? Do you still use the proposed parameterized family of coefficients $\gamma$ in this stage? If so, which parameters are set to random, and what distributions do they follow? How do you ensure the pre-training does not degrade performance much and effectively explores the parameter space?
>
> $\textbf{[3-1. Clarifying the design choice of the coefficient's hypothesis space section]}$
>
> Yes, we sample from the hypothesis space. We have revised numerous notations in Section 4.3 (Design Choice of the Coefficient's Hypothesis Space) to reduce ambiguity and enhance clarity. Specifically, the hypothesis space $\Gamma_h \subseteq \Gamma$ is defined as the set of $\gamma_\phi: [0, T] \times R^d \to R^{2 \times d}$, where:
>
> $\gamma_\phi = [\gamma_{0, \phi}, \gamma_{1, \phi}], \quad \phi \in P,$
>
> $\gamma_{0, \phi}(t, x_T) = T \frac{f_\phi(t, x_T)}{f_\phi(t, x_T) + g_\phi(t, x_T)}, \quad
> \gamma_{1, \phi}(t, x_T) = T \frac{g_\phi(t, x_T)}{f_\phi(t, x_T) + g_\phi(t, x_T)}.$
>
> $P$ represents the parameter space from which $\phi$ is drawn, and it determines the specific forms of $\gamma_{0, \phi}$ and $\gamma_{1, \phi}$ within $\Gamma_h$. The definitions of $f_\phi$ and $g_\phi$ are as follows:
>
> $f_\phi(t, x_T) = 1 - \frac{t}{T} + \left( \sum^M_{m=1} w^f_{m, \phi}(x_T) b_m(t) \right)^2, \quad
> g_\phi(t, x_T) = \frac{t}{T} + \left( \sum^M_{m=1} w^g_{m, \phi}(x_T) b_m(t) \right)^2,$
>
> where $w_\phi(x_T) = s \ \text{LPF} \circ \tanh \left( U_\phi(x_T) \right), \ s \in \mathbb{R}.$
>
> Random sampling from $\Gamma_h$ is performed as $w(u) = s \ \text{LPF} \circ u, \ u \sim \mathcal{N}(-1, 1) \in \mathbb{R}^{2 \times M \times d}.$
>
> $\textbf{[3-2. Ensuring the design choice of hypothesis space exclude crude coefficients]}$
>
> Thank you for pointing out this important aspect. We assume that trajectories including high frequencies in both $t$ and $d$ can be crude for transportation. To address this, we select sinusoidal functions with low frequencies and apply a Low-Pass Filter (LPF) across different dimensions $d$ to exclude high-frequency components.
>
> Specifically, during the pre-training stages, we tested various hyperparameters ($s$, $M$, and LPF configurations) and observed that performance slightly improved when multidimensionality near $t = T$ was reduced. For this purpose, we set the convolution group size for LPF to 1, resulting in an LPF output shape of $[B, 1, \text{resolution}, \text{resolution}]$. This constrains $\gamma$ to exhibit multidimensionality for small $t$ and reduced multidimensionality for large $t$. Details regarding this configuration are specifically illustrated in Appendix A due to space constraints. The empirical results for this are presented in Table 5. As shown, EDM$-\gamma$'s FID (2.08, 1.81) is nearly identical to EDM$-\alpha$'s FID (1.97, 1.79), indicating that our design choice of $\Gamma_h$ is empirically efficient.
>
> We acknowledge that this is not an optimal choice, and there remains significant room for improvement in the design, given that this is the beginning of the exploration for this topic.
>
> [$\textbf{Weakness 4.}$] The authors do not explain the design choice of the discriminator $D_\psi$ in Section 4.4 when it was first introduced.
>
> [$\textbf{Weakness 5.}$] Can you discuss the motivation for using hinge loss in the adversarial training in more detail? The authors just follow the previous work in GAN training without giving motivation.
>
> $\textbf{[4 and 5. Clarifying the discriminator and loss function for adversarial training]}$
>
> We employ the StyleGAN-XL [2] discriminator for $D_\psi$ with hinge loss [3], as used in CTM [4]. This choice is motivated by the near state-of-the-art performance of GAN-based methods utilizing the StyleGAN-XL architecture and loss function. Moreover, prior works on diffusion distillation, such as CTM, employed similar adversarial training setups, which informed our design. Reflecting all your feedback and comments, we revised the adversarial section with more details as shown below:
>
> "
> Following GAN [5], we use an adversarial loss to minimize Equation 6. Specifically, we employ the StyleGAN-XL [6] discriminator for $D_\psi$ with hinge loss [7] for $\theta$, $\phi$, and $\psi$ (discriminator parameters), as used in [4]. In our method, $\theta$ and $\phi$ in $G_{\theta, \phi}$ aim to deceive $D_\psi$. The loss function for $\theta$ is:
>
> $L_{\theta} = - E_{x_1 \sim \rho_1}[D_\psi \left( H_\theta(t, x(t), \gamma_\phi(t, z)) \right) ], \quad x(t) = x_0 + \gamma_{1, \phi}(t,z) \odot x_1, \quad z \sim \rho_T,$

---

> ### Author Response · Authors · 2024-11-17
>
> where $L_\theta$ indicates that the gradient is only calculated with respect to $\theta$. Since $\theta$ only needs to handle elements in $\{\gamma_\phi(t, x_T)}$ and not other elements in $\Gamma_h$, $\theta$ is trained exclusively on $\gamma_\phi$, helping reduce the load on $H_\theta$. The loss functions for $\phi$ and $\psi$ are:
>
> $L_\phi = - E_{x_1 \sim \rho_1}[D_\psi \left( G(\tau, x_1,  v_{\theta, \phi}) \right) ],$
>
> $L_\psi = E_{x_0 \sim \rho_0}[\max(0, 1 - D_\psi(x_0))] + E_{x_1 \sim \rho_1}[\max(0, 1 + D_\psi(G(\tau, x_1,  v_{\theta, \phi})))].$
>
> Here, $L_\phi$ and $L_\psi$ indicate that gradients are calculated with respect to $\phi$ and $\psi$, respectively, as shown in Figure 3. With these loss functions, $\phi$ is optimized to find better trajectories, while $\theta$ adapts to $\gamma_\phi$, which is sparser than $\Gamma_h$.
> "
>
> $[\textbf{Weakness 6.}]$ In Section 5.2, the authors mention that the NFE of the proposed method is 6 (+1 for $\gamma_\phi$). What exactly does this mean? Why “+1 for $\gamma_\phi$” rather than using learned $\gamma_\phi$ through the whole sampling?
>
> $\textbf{[6. Ambiguity of the 5 (+1) notation]}$
>
> There seems to be some misunderstanding in this part. We use learned $\gamma_\phi$ through the whole sampling. As the calculation of $\gamma_\phi$ involves function evaluation, we include this in the NFE count. To reduce ambiguity, we changed the notation from "5 (+1)" to "5 (+)" and added the clarification: "(+ for the calculation of $\gamma_\phi$, given that the network for $\gamma_\phi$ is smaller than the network for $H_\theta$)."
>
> [$\textbf{Weakness 7.}$] In Table 2-4, when comparing with other distillation methods, it would be more fair to run other methods under the same NFE. For example, how do CD and CTM perform with 5 or 6 NFE?
>
> $\textbf{[7. FID of CTM with 5 or 6 NFEs]}$
>
> We totally agree with your point. For a fair comparison with other distillation methods in terms of NFE, we select CTM [4] as a representative due to its popularity, high performance, and the use of the same model architecture based on EDM and adversarial training. We then calculate the FID using 5 and 6 NFE.
>
> CTM's FID for CIFAR-10 unconditional: 1.86 (NFE=5), 1.93 (NFE=6)
>
> CTM's FID for CIFAR-10 conditional: 1.98 (NFE=5), 2.04 (NFE=6)
>
> Ours achieves 1.69 (CIFAR-10 uncond) and 1.37 (SOTA on CIFAR-10 cond) with 5 NFE.
>
> As shown above, increasing the NFE of CTM does not significantly decrease the FID, and the FID even increases. This indicates that the adversarial approach for MTO provides additional performance gains that cannot be achieved by distillation methods alone, even with increased computational cost.
>
> [$\textbf{Weakness 8.}$] It would be great to add some comments in the caption of Figure 2 to help better explain the difference and illustrate the idea.
>
> $\textbf{[8. Adding more explanation for the figure]}$
>
> We completely agree with your suggestion. The caption has been updated to: "Crude coefficients: (a) Oscillatory behavior in $t$ due to high-frequency components; (b) High adjacent pixel differences in $d$. Refined coefficients: (c) Constrained multidimensionality for larger $t$ in pre-training; (d) Unconstrained multidimensionality for adversarial training." This should improve comprehension.
>
> Once again, we sincerely appreciate your detailed and specific feedback. We have uploaded a revised version of the PDF reflecting your comments as thoroughly as possible. If you have any further questions, we are happy to address them without hesitation.
>
> Thank you again.
>
> [1] Bowen Zheng and Tianming Yang. Diffusion models are innate one-step generators, 2024.
>
> [2] Axel Sauer, Katja Schwarz, and Andreas Geiger. Stylegan-xl: Scaling stylegan to large diverse
> datasets, 2022.
>
> [3] Jae Hyun Lim and Jong Chul Ye. Geometric gan, 2017.
>
> [4] Dongjun Kim, Chieh-Hsin Lai, Wei-Hsiang Liao, Naoki Murata, Yuhta Takida, Toshimitsu Uesaka, Yutong He, Yuki Mitsufuji, and Stefano Ermon. Consistency trajectory models: Learning probability flow ODE trajectory of diffusion, 2024.

---

> > ### Comment · Reviewer_xR7Z · 2024-11-25
> > **Reply to the rebuttal**
> >
> > I thank the authors for the detailed rebuttal.
> > The clarification helps me better understand the contribution.
> > The only concerns I have are: 1) one needs to train the model from scratch rather than leveraging existing models; 2) the learned coefficients seem to be quite restrictive (if we learn the coefficients for the 5-step sampler, we can not adapt it to a sampler with other NFEs.)
> >
> > Some typos still exist in the newly revised part of your draft.
> > For example, in Eq. (10), $x_1$ on the right-hand side should be $x_T$.
> > In the line 3 and the line 9 of Algorithm 1, $t \sim \mathcal{N}(-1.2, 1.2)$. My understanding is that $t \in (0, 1)$.

---

> > > ### Author Response · Authors · 2024-11-27
> > > **Extra experimentation results have been provided in the supplementary materials to validate our hypothesis space choice.**
> > >
> > > $\textbf{[3-2. Ensuring the design choice of hypothesis space excludes crude coefficients]}$
> > >
> > > We support our hypothesis space and hyperparameter choices for $s$, $M$, and $\sigma$ in the LPF through additional experimental results uploaded in the supplementary materials. These experiments were conducted using Stochastic Interpolants, Flow Matching, and Denoising Diffusion Probabilistic Models on image datasets (CIFAR-10 and ImageNet-32).
> > >
> > > As shown in Tables 1 and 2, we observed clear relationships between the pre-trained model's performance and the values of $s$, $M$, and $\sigma$. Three key findings emerged from these experiments:
> > >
> > > 1. Using LPF consistently outperforms non-LPF settings.
> > > 2. Ensuring sufficient $\sigma$ values for LPF significantly improves performance.
> > > 3. Selecting an appropriate (small) value for $M$ is critical for achieving optimal results.
> > >
> > > Based on these insights, we selected hyperparameters for pre-training and further experimented with MTO using different hyperparameter configurations, as presented in Table 6. These experiments allowed us to determine appropriate hyperparameter choices to optimize performance effectively.

---

> > > > ### Author Response · Authors · 2024-11-28
> > > > **Appendix E (Additional Experiment): Organizing and clarifying experiments in supplementary materials**
> > > >
> > > > We have added Appendix E: Additional Experiment in our main paper to organize and clarify the experiments presented in the supplementary materials. This section includes detailed descriptions of network architectures, training configurations, and concise discussions for each training result.
> > > >
> > > > We want to highlight three key points:
> > > >
> > > > 1. Pre-training with the multidimensional coefficient not only preserves but also enhances the performance of flow and diffusion models.
> > > > 2. All MTO experiments in Appendix E were conducted by training only $\phi$, while keeping $\theta$ frozen.
> > > > 3. Even with a small 4-layer discriminator network and Vanilla GAN Loss, we achieved comparable FID performance on SI.
> > > >
> > > > Thank you for your attention to these updates.

---

> > > > > ### Author Response · Authors · 2024-12-03
> > > > > **Summary of Revisions and Final Check**
> > > > >
> > > > > We have summarized our revisions and key points in a top comment for all reviewers. If there are any additional clarifications or points you’d like us to address before the discussion closes, please let us know. Thank you again for your thoughtful feedback.

---

> ### Author Response · Authors · 2024-11-26
> **Thanks reviewer xR7Z for the responses**
>
> First of all, thank you for your core and concise concerns, which are highly valuable for addressing the limitations of our paper and guiding follow-up research. We agree that the limitations you pointed out are important, and we have revised our limitations section to address these points.
>
> 1) $\textbf{One needs to train the model from scratch rather than leveraging existing models.}$
>
> Yes, the coefficient labeling (as described in Section 4.3) makes the model structure different from existing well pre-trained models like EDM. This requirement can indeed make applying MTO cumbersome. Future work could mitigate this issue by replacing a few layers from a well pre-trained model and using it as initialization for pre-training.
>
> 2) $\textbf{The learned coefficients seem to be quite restrictive (if we learn the coefficients for the 5-step sampler, we cannot adapt them to a sampler ...}$
>
> We agree with your point. Our current approach trades off flexibility for improved performance by restricting the inference sampling configurations. Future work could address this limitation by conditioning $\gamma_\phi$ on diverse sampling configurations, enabling better adaptability and efficiency.
>
> We have revised our limitation section as below:
>
> "First, since we use coefficient labeling (Section~4.3) for the diffusion model, the model structure differs from existing pre-trained flow and diffusion models. As a result, training models from scratch is required, which can be cumbersome. This issue could be mitigated in future works by replacing a few layers from well pre-trained model and using it as initialization for pre-training. Second, $\gamma_\phi$ is tied to the specific sampling configuration used during MTO, limiting its flexibility for inference under alternative configurations. Future work could address this by conditioning $\gamma_\phi$ on diverse sampling configurations, enabling better adaptability and efficiency. ...."
>
> 3) $\textbf{Some typos still exist in the newly revised part of your draft. For example, in Eq. (10), $x_1$ on the right-hand side ...}$
>
> Thank you for pointing out the typos we missed. We have revised Eq. (10) accordingly. Regarding $t \sim \mathcal{N}(-1.2, 1.2)$ in lines 3 and 9, this is actually correct given that $t \in [0, T]$ ($T = 80$ for EDM as in Equation 3), as stated in Definition 1 and 2. We will continue working diligently to ensure all typos are corrected in the final version.
>
> We have revised our PDF and uploaded it to reflect your feedback. Thank you once again for your constructive feedback and detailed observations.

---

### Official Review · Reviewer_EetL · 2024-11-04

**Soundness:** 3
**Presentation:** 3
**Contribution:** 2
**Rating:** 5
**Confidence:** 4

**Summary:**

This paper proposes a learning-based method to optimize the coefficients used in constructing the forward processes of flow- or diffusion-based models. When combined with an adversarial training objective, this method performs well on one benchmark dataset. The approach involves a two-stage process: pretraining followed by simulation-based tuning.

**Strengths:**

1. Extensive experiments were conducted on some of the latest models.
2. The method is easy to understand, and the writing is clear.

**Weaknesses:**

1. **Flow Matching Results**: If possible, please provide results on the image dataset based on the flow-matching model.

2. **Image Dataset Experiment with $N=5$**: In the image dataset experiment, $N=5$ was selected, seemingly for efficiency in image generation (simulating Equation (13)). Is this correct? Can a different sampler or time scheduler be used to draw samples after training?

3. **Ablation Study Findings**: The ablation study shows that EDM-$\gamma$ + Adv. $\theta$ achieves comparable performance. This adversarial objective was previously proposed and appears to be the critical component, rather than the proposed trajectory optimization coefficients.

4. **State-dependent Multidimensional Coefficients**: Can the multidimensional coefficients also depend on the state $x(t)$?

5. **Optimization of the $\gamma$ Model**: The $\gamma$ model is optimized only at specific time schedules (used in Equation (13)) during the second-stage training. Does this imply that the inference schedule used during training must match the schedule used during sampling?

6. **Training Efficiency in Simulation Dynamics**: Please include a discussion on the training efficiency.

**Writing Adjustments**:

- Verify the correctness of $\gamma_0(T)$ on the right-hand side of Equation (8).

**Questions:**

See above.

---

> ### Author Response · Authors · 2024-11-18
>
> First of all, we really appreciate your insights and valuable feedback. We are especially grateful for recognizing the following strengths of our work, which we summarize below:
>
> [$\textbf{STRENGTHS}$]
>
> We thank Reviewer EetL for their positive and insightful comments! Reviewer EetL noted the extensive experiments conducted in our work, highlighting the use of state-of-the-art models. We are encouraged that Reviewer EetL appreciated the comprehensiveness of our experimental validation, as it underscores the robustness of our methodology. Moreover, Reviewer EetL mentioned that our method is easy to understand and that the writing is clear. We are delighted that our efforts to present the ideas and technical details in an accessible manner were recognized. This feedback inspires us to continue ensuring clarity and accessibility in our work.
>
> [$\textbf{WEAKNESSES}$]
>
> [$\textbf{Weakness 1.}$] Flow Matching Results: If possible, please provide results on the image dataset based on the flow-matching model.
>
> [$\textbf{1. Adversarial approach to MTO for Flow Matching}$]
>
> It would be valuable to show empirical experiments with the FM (Flow Matching) [1] framework given its popularity. However, FM has limitations for performing MTO as it directly estimates the entire vector field without disentangling the coefficient and target distributions like EDM. In equation form, flow matching's objective is:
>
> $v(t, x(t)) \approx \hat{v}_\theta(t, x(t))$,
>
> while diffusion predicts the vector field by estimating noise and target distribution's samples $x_{0, \theta}$ and $x_{1, \theta}$ as follows:
>
> $v(t, x(t)) \approx \alpha_0^{'}(t)x_{0, \theta} + \alpha_1^{'}(t)x_{1, \theta}$.
>
> This difference makes it challenging to solve MTO in FM, as we must couple the vector field with $\theta$ and $\phi$, which is not feasible in FM. However, we can use SI (Stochastic Interpolant) [2] for MTO, as shown in our 2-dimensional experiments, given that it estimates $x_{0, \theta}$ and $x_{1, \theta}$ rather than directly modeling the vector field. SI is known for generalizing the flow and diffusion methodologies, which is why we employed SI in the 2-dimensional experiment to demonstrate the performance of a generalized FM methodology.
>
> [$\textbf{Weakness 2.}$] Image Dataset Experiment with $N=5$: In the image dataset experiment, $N=5$ was selected, seemingly for efficiency in image generation (simulating Equation (13)). Is this correct? Can a different sampler or time scheduler be used to draw samples after training?
>
> [$\textbf{Weakness 5.}$] Optimization of the $\gamma$ Model: The $\gamma$ model is optimized only at specific time schedules (used in Equation (13)) during the second-stage training. Does this imply that the inference schedule used during training must match the schedule used during sampling?
>
> [$\textbf{2 and 5. Answering questions for inference by using learned coefficient only for fixed sampling configurations}$]
>
> Thank you for pointing out these important properties of our method.
>
> First, you are correct. The reason for using NFE=5 in the image generation experiment was to prioritize training efficiency. We can employ various configurations by trading off computational cost and image quality.
>
> Second, since our training scheme depends on the sampling configuration, the inference schedule used during training must match the schedule used during sampling. Although we can utilize a different inference schedule, as $\gamma_\phi$ provides continuous coefficients, the outputs are not optimized for those schedules since $\gamma_\phi$ was not exposed to them during training.
>
> The problem we aim to solve is:
>
> “Given a differential equation solver $\textbf{with fixed configurations}$, which multidimensional trajectory yields optimal performance in terms of final transportation quality for a given starting point of the differential equation?”
>
> The additional performance improvements gained are derived from simulation-based objectives, which require fixed sampling strategies. We acknowledge that this can be a constraint for flexible inference and have revised the limitations section to address this topic as below:
>
> "solving MTO under fixed inference schedules may limit flexibility. Future research could explore general sampling configurations to enhance adaptability and efficiency."
>
> Thank you for your constructive feedback again.

---

> > ### Comment · Reviewer_EetL · 2024-11-25
> >
> > Thanks for your response.
> >
> > Q1: I still don't understand why the flow matching model is considered harder to tune using your method, while the stochastic interpolant is deemed feasible. I don't see any intrinsic difference between these two methods, especially since they are considered concurrent works on the same methodology. Moreover, papers explicitly using the term "stochastic interpolant" have demonstrated success in image generation tasks as well [1]. How can your methods be adapted to SI-based image generative models?
> >
> > Q5: When referring to "fixed configurations," what specific configurations are being discussed? Are they related to sampler hyperparameters, such as the number of steps for the Euler solver?
> >
> > Ref:\
> > [1] SiT: Exploring Flow and Diffusion-based Generative Models with Scalable Interpolant Transformers. ECCV 2024.

---

> > > ### Comment · Reviewer_EetL · 2024-11-25
> > >
> > > As reviewer ZX4x points out, the biggest technical concern with this paper is that the major performance gains are rooted in the GAN-based training objective. However, the "storytelling" in the paper does not clearly present it this way.
> > >
> > > Additionally, there are no interpretations provided for the learned coefficients generated by $\gamma_\phi$; these coefficients are purely determined by the noise vector at the initial denoising step, making them different across sampling runs.
> > >
> > > The current evidence does not support the claim that $\gamma_\phi$ learned in this manner is empirically or theoretically beneficial for real-world generative modeling tasks.
> > >
> > > I thank the authors for their efforts in responding, but I will maintain my score.

---

> ### Author Response · Authors · 2024-11-18
>
> [$\textbf{Weakness 3.}$] Ablation Study Findings: The ablation study shows that EDM-$\gamma$+ Adv achieves comparable performance. This adversarial objective was previously proposed and appears to be the critical component, rather than the proposed trajectory optimization coefficients.
>
> [$\textbf{3. Question about the practical benefits of optimizing $\gamma$ by observing ablation study}$]
>
> We agree with your point. It is true that adversarial training of $\theta$ is crucial for performance improvements. However, we emphasize that optimizing only $\gamma_\phi$ with a simulation-based end-to-end objective alone is effective and novel, as shown in the 2-dimensional experiment. These experiments validate the existence of performance gains achievable through optimizing only $\phi$. To clarify this, we added an explanation in 2-dimensional experiment section.
>
> Additionally, we highlight our training methodology by comparing it with CTM [3], which uses adversarial training for $\theta$ but not for $\phi$. For this comparison, we calculated FID using 5 and 6 NFE:
>
> CTM's FID for CIFAR-10 unconditional: 1.86 (NFE=5), 1.93 (NFE=6).
> CTM's FID for CIFAR-10 conditional: 1.98 (NFE=5), 2.04 (NFE=6).
> Ours achieves 1.69 (CIFAR-10 uncond) and 1.37 (SOTA on CIFAR-10 cond) with 5 NFE.
>
> As shown above, increasing the NFE of CTM does not significantly reduce FID, and it even increases slightly. This demonstrates that the adversarial approach for MTO provides performance gains that cannot be achieved by distillation methods alone, even with increased computational cost. These performance improvements for near-SOTA and SOTA results stem from our unique training methodology focusing on jointly training $\theta$ and $\phi$.
>
> We also highlight that training $\theta$ and $\phi$ using simulation-based end-to-end optimality is typically inefficient, but our methodology addresses this issue. This point is elaborated further in [$\textbf{6. Training efficiency of simulation-dynamics}$].
>
> By these additional experiments and observations, we emphasize two key aspects:
>
> 1. The effectiveness of flexibility and adaptability achieved through the adaptive multidimensional coefficient $\gamma_\phi$.
> 2. The practical benefits of combining simulation-based end-to-end optimality with simulation-free methodologies, leveraging the advantages of each while mitigating their limitations.
>
> These comparisons and insights underscore the practical benefits of adversarial approach to MTO. Thank you again for pointing out this critical aspect.
>
> [$\textbf{Weakness 4.}$] State-dependent Multidimensional Coefficients: Can the multidimensional coefficients also depend on the state?
>
> [$\textbf{4. State-dependent adaptive multidimensional coefficient}$]
>
> Yes, you are absolutely correct. After reading your comments, we realized it would be beneficial to define state-dependent multidimensional coefficients separately and clarify our choice. For this purpose, we introduced a new definition called the adaptive multidimensional coefficient, which captures the concept of adaptability for various inference trajectories. The definition is as follows:
>
> $\textbf{Definition 2: Adaptive Multidimensional Coefficient}$
> For $t \in [0, T]$ and inference trajectory $x_{\theta, \phi}(t)$, the adaptive multidimensional coefficient $\gamma_\phi(t, x_{\theta, \phi}(t)):[0, T] \times \mathbb{R}^d \rightarrow \mathbb{R}^{2 \times d}$ is parameterized by $\phi$. Boundary conditions follow Definition 1 (Multidimensional Coefficient), with $\gamma_\phi \in C^1([0, T], \mathbb{R}^{2 \times d})$.
>
> This definition describes the trajectory-conditioned multidimensional coefficient. To reduce computational costs in calculating $\gamma_\phi$, we use only $t = T$ for $x_{\theta, \phi}(t)$ in $\gamma_\phi(t, x_{\theta, \phi}(t))$ rather than the inference trajectory at multiple time points. This approach allows us to compute $\gamma_\phi$ across the entire inference time schedule $\tau = \{ t_0, \ldots, t_N \}$ with a single function evaluation before initiating transportation. By using the adaptive multidimensional coefficient, we can address multidimensional trajectory optimization more rigorously.
>
> Thank you for your insightful suggestion, which has helped us better articulate this important concept.

---

> ### Author Response · Authors · 2024-11-18
>
> [$\textbf{Weakness 6.}$] Training Efficiency in Simulation Dynamics: Please include a discussion on the training efficiency.
>
> [$\textbf{6. Training efficiency of simulation-dynamics}$]
>
> We found that the training efficiency of our methodology is significantly better than other distillation methods, such as GDD [4]. We evaluated the number of images required for training and the training time, comparing these with GDD and GDD-I [4]. Remarkably, our approach for MTO requires far fewer images for training across all datasets compared to GDD-I, a method known for its efficiency in diffusion distillation:
>
> CIFAR-10: 5000kimg (GDD-I) vs. 1382kimg (Ours)
> FFHQ: 10000kimg (GDD-I) vs. 106kimg (Ours)
> AFHQv2: 10000kimg (GDD-I) vs. 955kimg (Ours)
>
> The training times for our method are 10, 2, and 6 hours for CIFAR-10, FFHQ, and AFHQv2, respectively, which are also comparatively low. Additionally, FID significantly decreases in the early stages of training, as illustrated in Figure 6 (we modified the x-axis from iteration number to kimg for improved readability). The primary computational cost arises from VRAM requirements; however, training remains practically feasible. Good performance is achieved with just 5 NFE, which is relatively low. All experiments for adversarial training were conducted on GPUs with 48GB of VRAM—less than the 80GB VRAM GPUs often used in related works.
>
> These results underscore the effectiveness of simulation-based end-to-end optimality while highlighting the strength of combining simulation-free and simulation-based methodologies, leveraging the advantages of each while mitigating their limitations. Considering these aspects, we estimate that our adversarial approach for MTO is scalable to larger datasets while maintaining efficiency.
>
> [$\textbf{Weakness 7.}$] Writing adjustments: Verify the correctness of $\gamma_0(T)$ on the right-hand side of Equation (8).
>
> [$\textbf{7. Writing adjustments for boundary conditions}$]
>
> Yes, $\gamma_0(T)$ on the right-hand side of Equation (8) is correct. For cases like $T=1$, $k$ should be $k=0$ to satisfy the boundary conditions. Boundary conditions like this ensure that $x(t)$ becomes $x_0$ and $x_T$ for $t=0$ and $t=T$, which is a requirement for transportation. The values of $k$ and $T$ for boundary conditions vary based on the task. To avoid misunderstandings, we have added an additional explanation of the boundary conditions for better comprehension. Thank you for your observation.
>
> We sincerely thank you once again for your thoughtful feedback and valuable insights. Your suggestions have been instrumental in helping us clarify and enhance the presentation, scalability, and technical rigor of our work. We have carefully revised the paper to address all the comments and uploaded a new version that incorporates these changes.
>
> If you have any additional questions or further comments, we would be more than happy to address them. Thank you once again for your constructive and encouraging feedback.
>
> [1] Yaron Lipman, Ricky T. Q. Chen, Heli Ben-Hamu, Maximilian Nickel, Matt Le, Flow Matching for Generative Modeling, 2023.
>
> [2] Michael S Albergo, Nicholas M Boffi, and Eric Vanden-Eijnden. Stochastic interpolants: A unifying
> framework for flows and diffusions, 2023
>
> [3] Dongjun Kim, Chieh-Hsin Lai, Wei-Hsiang Liao, Naoki Murata, Yuhta Takida, Toshimitsu Uesaka, Yutong He, Yuki Mitsufuji, and Stefano Ermon. Consistency Trajectory Models: Learning Probability Flow ODE Trajectory of Diffusion, 2024.
>
> [4] Bowen Zheng and Tianming Yang. Diffusion models are innate one-step generators, 2024.

---

> ### Author Response · Authors · 2024-11-26
> **Thanks for Reviewer EetL for the response (Extra experimentation results are uploaded in supplementary materials)**
>
> First of all, thank you for your valuable questions. We have uploaded the extra experimental results you requested in the supplementary materials.
>
> $\textbf{Q1-1. I still don't understand why the flow matching model is considered harder to tune using your method, while the stochastic interpolant is ...}$
>
> We first want to point out that for MTO, it is essential to know each $\hat{x}_0$ and $\hat{x}_1$ separately. The reason why it is challenging to perform MTO on FM lies in the target value of the FM model. All three methods (SI, EDM, and Flow Matching) target different values, as shown below:
>
> 1. Flow Matching: $H_\theta(t, x(t)) = \hat{v} \approx v = \dot{\gamma}_0 x_0 - \dot{\gamma}_1 x_T$
> 2. SI: $H_\theta(t, x(t)) = [\hat{x}_0, \hat{x}_1] \approx [x_0, x_1]$
> 3. EDM (Diffusion): $H_\theta(t, x(t)) = \hat{x}_0 \approx x_0$
>
> For SI and EDM, the model's target value is disentangled from its time schedules. Thus, we can directly compute $[\hat{x}_0, \hat{x}_1]$ (for EDM, $x_1 = \frac{x(t) - \hat{x}_0}{\gamma_1}$) without dependency on the coefficients $\gamma$. This makes MTO feasible, as we can explicitly control $\gamma$ using $\phi$ during the sampling stage.
>
> However, in FM, the model's target value is entangled with the coefficients $\gamma$, as seen above. To calculate $\hat{x}_0$ and $\hat{x}_1$ for MTO in FM, unlike SI or EDM, solving the following simultaneous equations is required, which is computationally onerous:
>
> $H_\theta(t, x(t)) = \dot{\gamma}_0 x_0 + \dot{\gamma}_1 x_1$
>
> $x(t) = \gamma_0 x_0 + \gamma_1 x_1$
>
> Solving these simultaneous equations requires the continuous derivative values of coefficients $\dot{\gamma}$. However, using both $\dot{\gamma}$ and $\gamma$ values is practically inefficient and can confuse $\gamma_\phi$, as numerical errors are inevitable when sampling with a limited number of steps. In other words:
>
> $\dot{\gamma}_\phi(t_i) \neq \Delta\gamma(t_i) = \gamma(t_j) - \gamma(t_i) \quad (j = i+1)$
>
> Given the above equations, we only use $\gamma$ and not $\dot{\gamma}$ for SI and EDM (as detailed in Appendix B). However, to achieve this in FM, onerous computation is required to calculate $\Delta t_i v_{\theta, \phi}$ while avoiding the use of $\dot{\gamma}$. Nevertheless, \$\textbf{we performed MTO in FM on CIFAR-10.}$ You can see these results in supplementary material Tables 3, 4, and 5. As shown, the performance of FM is consistently worse than SI. This aligns with the reasoning provided above:
>
> 1. FM's model output is not disentangled from $\dot{\gamma}_\phi$.
> 2. Onerous computational steps are required for MTO in FM to calculate $\Delta t_i v_{\theta, \phi}$, which is computationally inefficient and can accumulate numerical errors.
>
> $\textbf{Q1-2. How can your methods be adapted to SI-based image generative models?}$
>
> It is entirely feasible to train SI-based image generative models. The only difference between SI and EDM lies in their target values and parameterization. We performed image generation experiments with SI, but it performed worse than EDM (as shown in supplementary material Table 3). As EDM is known for its high performance due to its well-tuned parameterization, we focused on experimenting with EDM to validate the empirical performance of MTO. We believe this sufficiently demonstrates the validity of our methodology by achieving on-par performance compared to other few-step generation methods.
>
> $\textbf{Q5: When referring to "fixed configurations," what specific configurations are being discussed? Are they related to sampler hyperparameters ...}$
>
> Yes, specifically, the time schedule $\tau = \{t_0, \dots, t_N\}$ for inference and the solving methodology (e.g., Euler, 2nd-order Heun) are the fixed configurations being discussed. This can be freely adjusted, as shown in Table 4 of the supplementary materials. We experimented with different NFEs for SI, FM, and DDPM.

---

> ### Author Response · Authors · 2024-11-26
> **Thanks for Reviewer EetL for the response (Extra experimentation results are uploaded in supplementary materials)**
>
> $\textbf{Q. The current evidence does not support the claim that $\gamma_\phi$ learned in this manner is empirically or theoretically beneficial for real-world ...}$
>
> We have uploaded additional experimental results to support the usage of $\gamma_\phi$ determined by the noise vector $x_T$. Specifically, we experimented with three different inputs for $\gamma_\phi$:
>
> 1. a vector filled with $1$
> 2. a random vector $z$ sampled from $\rho_T$ but unrelated to $x_T$
> 3. $x_T$.
>
> As shown in Table 5 of the supplementary material, using $x_T$ as the input for $\gamma_\phi$ consistently achieves better performance across three different methodologies. This provides empirical evidence supporting the advantage of the "Adaptive Multidimensional Coefficient" over using the same coefficient for all different $x_T$.
>
> Thank you for pointing out the missing logic in our paper, which we have now clarified.

---

> ### Author Response · Authors · 2024-11-27
> **Thanks for Reviewer EetL for the response (Extra experimentation results are uploaded in supplementary materials)**
>
> $\textbf{Q. As reviewer ZX4x points out, the biggest technical concern with this paper is that the major performance gains are rooted ...}$
>
> We acknowledge that the loss function itself is not highly novel. However, $\textbf{our approach is non-trivial:}$ using distinct objectives for $\theta$ and $\phi$, achieving remarkable efficiency and performance. Also, we would like to emphasize the contribution of the additional $\phi$ parameterization and its optimization, which is supported by extra experimental results in the supplementary materials.
>
> We want to highlight that all the experiments in our supplementary materials for MTO are $\textbf{obtained by training only $\phi$, while keeping $\theta$ frozen}$. As shown in the supplementary materials, particularly in the SI results on CIFAR-10 and ImageNet-32, we achieved comparably good FID values using only a few steps. These additional experimental results demonstrate that SI and MTO are a good combination, potentially sparking new interest in SI methodology. While these FID values are not close to those of current SOTA methodologies, this is due to the limitations of FM and SI and optimizing only $\phi$ rather than limitations of MTO itself, given that we demonstrated SOTA performance (CIFAR-10 conditional) using EDM in the main paper.
>
> Overall, the results in the supplementary materials show that we achieved on-par performance by optimizing only $\phi$ with an adversarial objective across various methodologies. This underscores the novelty and effectiveness of our approach in leveraging $\phi$ optimization for trajectory adaptability and improved generative performance.
>
> Thank you for your valuable responses.

---

> ### Author Response · Authors · 2024-11-28
> **Appendix E (Additional Experiment): Organizing and clarifying experiments in supplementary materials**
>
> We have added Appendix E: Additional Experiment in our main paper to organize and clarify the experiments presented in the supplementary materials. This section includes detailed descriptions of network architectures, training configurations, and concise discussions for each training result.
>
> We want to highlight three key points:
>
> 1. Pre-training with the multidimensional coefficient not only preserves but also enhances the performance of flow and diffusion models.
> 2. All MTO experiments in Appendix E were conducted by training only $\phi$, while keeping $\theta$ frozen.
> 3. Even with a small 4-layer discriminator network and Vanilla GAN Loss, we achieved comparable FID performance on SI.
>
> This result addresses your concern that "major performance gains are rooted in the GAN-based training objective." Additionally, we have revised the adversarial training section and clarified the overall explanation throughout the paper.
>
> Thank you for your attention to these updates.

---

> > ### Author Response · Authors · 2024-12-03
> > **Summary of Revisions and Final Check**
> >
> > We have summarized our revisions and key points in a top comment for all reviewers. If there are any additional clarifications or points you’d like us to address before the discussion closes, please let us know. Thank you again for your thoughtful feedback.

---

### Official Review · Reviewer_ZX4x · 2024-11-05

**Soundness:** 2
**Presentation:** 3
**Contribution:** 2
**Rating:** 6
**Confidence:** 3

**Summary:**

The paper presents a novel approach to trajectory optimization in generative models by introducing Multidimensional Trajectory Optimization (MTO). This method extends traditional flow and diffusion models by employing multidimensional coefficients, optimizing the trajectories of differential equations based on final sample quality rather than predefined trajectory shapes. The authors combine this with adversarial training to improve generative performance. The approach is validated across several datasets and generative models, with the results showing improvements in performance (e.g., CIFAR-10) with low function evaluations (NFEs). The proposed method outperforms or matches the state-of-the-art models in terms of quality, with a focus on optimizing the efficiency of the inference process.

**Strengths:**

**1. Interesting Introductions of the Multidimensional Coefficient**: The core innovation of the paper lies in the introduction of a multidimensional coefficient for trajectory optimization. This new formulation allows for more flexible and adaptive trajectories in flow and diffusion-based generative models, potentially improving generative model performance in a way that traditional unidimensional coefficients cannot.

**2. Comprehensive Empirical Validation**: The authors conduct extensive experiments on both synthetic 2D datasets and high-dimensional datasets like CIFAR-10, FFHQ, and AFHQv2. The experimental results demonstrate that MTO provides clear performance improvements, particularly with lower NFE, showcasing its practical benefits for high-quality sample generation.

**Weaknesses:**

**1. Incremental Contribution and Lack of Novelty**: While the multidimensional coefficient is an interesting extension of traditional trajectory optimization, adversarial training—the method used to optimize these trajectories—has already been explored in previous works (e.g., Adversarial Diffusion Distillation and Consistency Trajectory Models). The primary novelty in this work lies in the multidimensional coefficient rather than the adversarial training component itself. However, after a careful reading of the paper, I think the few-step generation with a multidimensional coefficient seems like a direct generalization of single-step generations such as CTM, UFOGen, etc, with incremental contributions. Another weakness is that the similar one-step generative model such as GDD and GDD-I on CIFAR10-unsound in Table 2, and results in Tables 3 and 4 outperform MTO models with up to 5 generation steps. This makes the contribution and practical performances of MTO questionable. The authors should more explicitly highlight how their method differs from and improves upon these existing adversarial training approaches. A more thorough discussion of this relationship would clarify the unique contributions of this paper.

**2. Clarity and Accessibility**: Some of the mathematical formulations, particularly those related to the multidimensional coefficients and their integration into the trajectory optimization process, may be challenging for readers unfamiliar with the specifics of generative models or differential equations. Including additional illustrations or intuitive explanations could improve the accessibility of the work. Additionally, clearer descriptions of the adversarial loss function and its interaction with the multidimensional coefficients would help readers better understand the core mechanics of the method.

**3. Limitations and Theoretical Insights**: The paper briefly mentions the lack of a theoretical foundation for the effectiveness of MTO. This could be addressed by offering some initial theoretical insights. Discussing the computational complexity of training models with multidimensional coefficients and potential limitations in scalability would also be valuable.

**4. Insufficiently related works**: Another weakness of the paper is the insufficient presentation of related works. The few and one-step diffusion distillation is a pretty mature research direction. However, the author fails to mention a lot of well-established approaches such as DMD [1], UFOGen [2], Diffusion-GAN [3], Diff-Instruct [4], iCT [5], GET-onestep [6], TRACT [7], etc. Besides, the authors compare MTO   on FFHQ and AFHQ datasets with a limited comparison with broader domains of other few-step generative models. I am curious why the authors do not conduct experiments on the well-established ImageNet-64 generation benchmark.

[1] One-step diffusion with distribution matching distillation

[2] UFOGen: You Forward Once Large Scale Text-to-Image Generation via Diffusion GANs

[3] Diffusion-GAN: Training GANs with Diffusion

[4] Diff-Instruct: A Universal Approach for Transferring Knowledge From Pre-trained Diffusion Models

[5] Improved Techniques for Training Consistency Models

[6]  One-step diffusion distillation via deep equilibrium models

[7] Tract: Denoising diffusion models with transitive closure time-distillation

**Questions:**

**1.** What is the intuition of the $L(\theta)$ objective in Equation (14).

**2.** I think it would be good if the authors could show the performances on the ImageNet64 benchmark.

**3.** Is MTO a direct generation of diffusion-induced GAN? What is the technical novelty behind MTO?

---

> ### Author Response · Authors · 2024-11-17
>
> First of all, we really appreciate your insights and valuable feedback. We are especially grateful for recognizing the following strengths of our work, which we summarize below:
>
> [$\textbf{STRENGTHS}$]
>
> We thank Reviewer ZX4x for thoughtful and positive feedback! We are encouraged that Reviewer ZX4x found our introduction of the multidimensional coefficient to be a novel and impactful core innovation. Reviewer ZX4x appreciates how this new formulation allows for more flexible and adaptive trajectories in flow and diffusion-based generative models, potentially leading to significant performance improvements over traditional unidimensional coefficients. We are glad that our work was recognized as a meaningful contribution to trajectory optimization. Reviewer ZX4x also highlighted the comprehensiveness of our empirical validation. Reviewer ZX4x noted that our extensive experiments on both synthetic 2D datasets and high-dimensional datasets like CIFAR-10, FFHQ, and AFHQv2 demonstrated clear performance improvements. Particularly, Reviewer ZX4x emphasized the practical benefits of MTO, such as its ability to generate high-quality samples with lower NFE. We are pleased that Reviewer ZX4x acknowledges the practical advantages of our approach.
>
> Below, we address the weaknesses mentioned in your comments in detail:
>
> [$\textbf{WEAKNESSES}$]
>
> [$\textbf{Weakness 1.}$] Incremental Contribution and Lack of Novelty: While the multidimensional coefficient is an interesting extension of traditional trajectory optimization, adversarial training—the method used to optimize these trajectories—has already been explored in previous works (e.g., Adversarial Diffusion Distillation and Consistency Trajectory Models). The primary novelty in this work lies in the multidimensional coefficient rather than the adversarial training component itself. However, after a careful reading of the paper, I think the few-step generation with a multidimensional coefficient seems like a direct generalization of single-step generations such as CTM, UFOGen, etc, with incremental contributions. Another weakness is that the similar one-step generative model such as GDD and GDD-I on CIFAR10-unsound in Table 2, and results in Tables 3 and 4 outperform MTO models with up to 5 generation steps. This makes the contribution and practical performances of MTO questionable. The authors should more explicitly highlight how their method differs from and improves upon these existing adversarial training approaches. A more thorough discussion of this relationship would clarify the unique contributions of this paper.
>
> $\textbf{[1-1. Comparing more rigorously with other methods with additional definition]}$
>
> After reading your comments, we realized that we didn't compare our methodology to other existing research thoroughly. We apologize for the poor explanations. Based on this, we have added and clarified the definitions in our methods for better comparison.
>
> We introduce a new definition called the adaptive multidimensional coefficient, which captures the concept of adaptability for various inference trajectories. The definition is written below:
>
> $\textbf{Definition 2: Adaptive Multidimensional Coefficient}$
> For $t \in [0, T]$ and inference trajectory $x_{\theta, \phi}(t)$, the adaptive multidimensional coefficient $\gamma_\phi(t, x_{\theta, \phi}(t)):[0, T] \times \mathbb{R}^d \rightarrow \mathbb{R}^{2 \times d}$ is parameterized by $\phi$. Boundary conditions follow Definition 1 (Multidimensional Coefficient), with $\gamma_\phi \in C^1([0, T], \mathbb{R}^{2 \times d})$.
>
> This definition describes the trajectory-conditioned multidimensional coefficient. To reduce computational costs in calculating $\gamma_\phi$, we use only $t = T$ for $x_{\theta, \phi}(t)$ in $\gamma_\phi(t, x_{\theta, \phi}(t))$ rather than the inference trajectory at multiple time points. This approach allows us to compute $\gamma_\phi$ across the entire inference time schedule $\tau = \{ t_0, \ldots, t_N \}$ with a single function evaluation before initiating transportation. By using the adaptive multidimensional coefficient, we can address multidimensional trajectory optimization more rigorously.
>
> By introducing this additional definition, we can now compare MTO and our adversarial approach to solve MTO with other research in terms of four major perspectives listed below:
>
> (i) Does $\phi$ exhibit multidimensionality? (multidimensional coefficient)
>
> (ii) Is $\phi$ trajectory-dependent? (adaptive multidimensional coefficient defined above)
>
> (iii) Are $\theta$ and $\phi$ jointly learnable?
>
> (iv) Is optimality defined by the final transportation quality?

---

> ### Author Response · Authors · 2024-11-17
>
> Upon comparing existing distillation methods for diffusions, we found that most distillation methods don't satisfy all four properties listed above. First, existing work doesn't use the multidimensional coefficient (i) or adaptive multidimensional coefficient (ii), as these concepts are introduced in our paper. Second, rather than optimizing $\theta$ and $\phi$ together (iii), existing distillation work only optimizes $\theta$. Since the trajectory for flow and diffusion $x_{\theta, \phi}(t) = \gamma_{0, \phi}(t)x_{0, \theta} + \gamma_{1, \phi}(t)x_{1, \theta}$ is defined by both flow and diffusion parameters $\theta$ and coefficient parameters $\phi$, distillation methodologies lack full flexibility for the trajectory, so they cannot be truly called "trajectory optimization." Lastly, diffusion distillation methods typically follow a target teacher network, meaning optimality cannot solely be evaluated by the quality of the final transportation, which is the key measure of generative modeling.
>
> For a more rigorous comparison, we examined additional related works in the trajectory optimization section, including WTH ("where to diffuse, how to diffuse, and how to get back") [1] and NFDM ("Neural flow diffusion models: Learnable forward process for improved diffusion modeling") [2]. [1] defines optimality through a fixed sequence of diffusion steps aimed at reducing inference complexity rather than focusing on the quality of final samples. Additionally, [2] introduces neural flow models that implicitly set trajectory optimality within the diffusion process, aiming to generate high-quality samples without explicit trajectory adjustment post-training. While these methods satisfy (iii) and have diverse perspectives on trajectory optimality, there are two significant differences between these methods and ours. First, we calculate the trajectory's optimality solely based on the final transportation quality (iv), which is crucial in generative modeling. Second, none of these methods achieve full trajectory flexibility regarding multidimensionality (i) and adaptability (ii) for different inference trajectories.
>
> Comparing more broadly with the related works you mentioned and additional works we identified was highly beneficial in clarifying our methodology. We apologize again for the previous poor explanation, and thank you for your specific and kind identification of related works and comments. We have revised the related work section as written below:

---

> ### Author Response · Authors · 2024-11-17
>
> "
> $\textbf{Trajectory Optimizations in Flow and Diffusion}$
> Various trajectory optimization approaches pre-define optimality without relying on final transportation quality. Approaches such as
> [9, 10] define straightness as the optimality criterion and optimize trajectories by maintaining the consistency of $(x_0, x_1)$ for training flow and diffusion models, aligning with an optimal transport perspective. Another example is [1], which defines optimality through a fixed sequence of diffusion steps intended to reduce inference complexity rather than focusing on the quality of final samples. Additionally, [2] introduces neural flow models that implicitly set trajectory optimality within the diffusion process, aiming to generate high-quality samples without explicit trajectory adjustment post-training. There are also approaches that refine trajectories after training, such as [11], where optimality is defined by minimizing the trajectory length in the Wasserstein-2 metric, focusing on a shortest-distance criterion. Despite these diverse perspectives on trajectory optimality, there are two significant differences between these methods and ours. First, we calculate the optimality of the trajectory solely based on the final transportation quality, which is a crucial factor in generative modeling. Second, none of these methods achieve full flexibility of the trajectory on two fronts: multidimensionality and adaptability with respect to different inference trajectories.
>
> $\textbf{Diffusion Distillation for Few-Step Generation}$
> There are two main approaches to diffusion distillation: non-adversarial and adversarial. Non-adversarial methods, like [12, 16, 17, 18], focus on 1-step distillation techniques without adversarial objectives. These approaches aim to simplify training by leveraging distributional losses and equilibrium models, effectively distilling the diffusion process without involving a GAN discriminator. Conversely, adversarial approaches to diffusion distillation, such as [3, 13, 14], employ a GAN-like discriminator to enhance sample quality by learning distribution consistency in an adversarial setting. Additionally, [15] propose Diff-Instruct, which transfers knowledge from pre-trained diffusion models through a GAN-based framework, closely resembling GAN approaches in training dynamics. Also, [4] developed Consistency Trajectory Models (CTM), which generalize [19] for efficient sampling with the assistance of a discriminator. While our method shares similarities in achieving few-step generation, there is a key difference: distillation in these works is not aimed at trajectory optimization. Our method optimizes both $\theta$ and $\phi$, representing the diffusion model parameters and multidimensional coefficient parameters, with respect to different inference trajectories, thereby enabling adaptive multidimensional trajectory optimization. In contrast, distillation typically targets a fixed teacher network, limiting flexibility in optimizing the trajectory.
> "
>
> Overall, this work achieves full trajectory flexibility and adaptability through end-to-end adversarial training—previously only achievable with the high training costs of simulation-based objectives—while preserving training efficiency, as detailed in next section.
>
> $\textbf{[1-2. Highlighting the practical benefits of the adversarial approach to MTO with additional experiments and observations]}$
>
> Thank you for pointing out the practical concernable points of MTO. Our explanation for comparing practical aspects like training efficiency and fair comparison with other methods was lacking. To address this, we conducted additional experimentation and observations.
>
> First, the training efficiency of our methodology is significantly better than other distillation methods like GDD [3]. We evaluated the number of images required for training and the training time, comparing these with GDD and GDD-I [3]. Remarkably, our approach for MTO requires far fewer images for training across all datasets compared to GDD-I, a method known for its efficiency in diffusion distillation:
>
> CIFAR-10: 5000kimg (GDD-I) vs. 1382kimg (Ours)
>
> FFHQ: 10000kimg (GDD-I) vs. 106kimg (Ours)
>
> AFHQv2: 10000kimg (GDD-I) vs. 955kimg (Ours)

---

> ### Author Response · Authors · 2024-11-17
>
> The training times for our method are 10, 2, and 6 hours for CIFAR-10, FFHQ, and AFHQv2, respectively, which are also comparatively low. Additionally, FID significantly decreases in the early stages of training, as illustrated in Figure 6 (we modified the x-axis from iteration number to kimg for improved readability). The primary computational cost arises from VRAM requirements; however, training remains practically feasible. Good performance is achieved with just 5 NFE, which is relatively low. All experiments for adversarial training were conducted on GPUs with 48GB of VRAM—less than the 80GB VRAM GPUs often used in related works. These results underscore the effectiveness of simulation-based end-to-end optimality while highlighting the strength of combining simulation-free and simulation-based methodologies, leveraging the advantages of each while mitigating their limitations. Considering these aspects, we estimate that our adversarial approach for MTO is scalable to larger datasets while maintaining efficiency. We have added a new paragraph like this in the experiments section to address these scalability issues.
>
> Second, for a fair comparison with other distillation methods in terms of NFE, we select CTM [4] as a representative due to its popularity, high performance, and the use of the same model architecture based on EDM and adversarial training. We then calculate the FID using 5 and 6 NFE:
>
> CTM's FID for CIFAR-10 unconditional: 1.86 (NFE=5), 1.93 (NFE=6)
>
> CTM's FID for CIFAR-10 conditional: 1.98 (NFE=5), 2.04 (NFE=6)
>
> Ours achieves 1.69 (CIFAR-10 uncond) and 1.37 (SOTA on CIFAR-10 cond) with 5 NFE.
>
> As shown above, increasing the NFE of CTM does not significantly decrease the FID, and the FID even increases. This indicates that the adversarial approach for MTO provides additional performance gains that cannot be achieved by CTM alone, even with increased computational cost.
>
> Lastly, we want to highlight the strong potential of the adversarial approach to MTO in conditional generation settings based on additional observations. We observed that $w_\phi$ exhibits very similar values for the same label condition. This implies that in conditional settings, we can significantly reduce the burden on $H_\theta$ for learning diverse $\gamma_\phi$, enabling us to achieve SOTA results. We updated our Figure 9 with labels for better readability.
>
> By considering these perspectives of MTO and the adversarial approach to MTO, we aim to demonstrate that our methods have substantial practical benefits.
>
> [$\textbf{Weakness 2.}$] Clarity and Accessibility: Some of the mathematical formulations, particularly those related to the multidimensional coefficients and their integration into the trajectory optimization process, may be challenging for readers unfamiliar with the specifics of generative models or differential equations. Including additional illustrations or intuitive explanations could improve the accessibility of the work. Additionally, clearer descriptions of the adversarial loss function and its interaction with the multidimensional coefficients would help readers better understand the core mechanics of the method.
>
> $\textbf{[2. Improving clarity and accessibility]}$
>
> We apologize for the poor illustrations in the paper. We completely agree with your comments and have worked on improving clarity and accessibility in various sections of our paper.
>
> First, we added an additional explanation for the boundary condition of the definition of the multidimensional coefficient. Specifically, we included the sentence: "Boundary conditions written above ensure that $x(t)$ becomes $x_0$ and $x_T$ for $t=0$ and $t=T$, which is a requirement for transportation. Values $k$ and $T$ for boundary conditions vary based on the task." Additionally, we introduced the definition of the 'adaptive multidimensional coefficient' as detailed above in the "[1-1. Comparing more rigorously with other methods with additional definition]" section of the comment for better understanding.
>
> To clarify trajectory optimization, we added the sentence: "Given that $\textbf{the trajectory}$ $x_{\theta, \phi}(t)$ from $G_{\theta, \phi}$ is entirely parameterized by $\theta$ and $\phi$, we have full controllability over the trajectory by adjusting $\theta$ and $\phi$, allowing us to term this process a $\textbf{trajectory optimization}$." This is included after the definition of trajectory optimization to help readers understand that trajectory optimization involves both coefficient $\phi$ and flow or diffusion model parameters $\theta$.

---

> ### Author Response · Authors · 2024-11-17
>
> We also upgraded the clarity of the "design space for hypothesis space" section by using $\Gamma_h$ as our notation for the hypothesis space. Specifically, the hypothesis space $\Gamma_h \subseteq \Gamma$ is defined as the set of $\gamma_\phi: [0, T] \times R^d \to R^{2 \times d}$, where:
>
> $\gamma_\phi = [\gamma_{0, \phi}, \gamma_{1, \phi}], \quad \phi \in P,$
>
> $\gamma_{0, \phi}(t, x_T) = T \frac{f_\phi(t, x_T)}{f_\phi(t, x_T) + g_\phi(t, x_T)}, \quad
> \gamma_{1, \phi}(t, x_T) = T \frac{g_\phi(t, x_T)}{f_\phi(t, x_T) + g_\phi(t, x_T)}.$
>
> $P$ represents the parameter space from which $\phi$ is drawn, and it determines the specific forms of $\gamma_{0, \phi}$ and $\gamma_{1, \phi}$ within $\Gamma_h$. The definitions of $f_\phi$ and $g_\phi$ are as follows:
>
> $f_\phi(t, x_T) = 1 - \frac{t}{T} + \left( \sum^M_{m=1} w^f_{m, \phi}(x_T) b_m(t) \right)^2, \quad
> g_\phi(t, x_T) = \frac{t}{T} + \left( \sum^M_{m=1} w^g_{m, \phi}(x_T) b_m(t) \right)^2,$
>
> where $w_\phi(x_T) = s \ \text{LPF} \circ \tanh \left( U_\phi(x_T) \right), \ s \in \mathbb{R}.$
>
> For the adversarial training section, we revised the content as follows for clarity:
>
> "
> Following GAN [5], we use an adversarial loss to minimize Equation 6. Specifically, we employ the StyleGAN-XL [6] discriminator for $D_\psi$ with hinge loss [7] for $\theta$, $\phi$, and $\psi$ (discriminator parameters), as used in [4]. In our method, $\theta$ and $\phi$ in $G_{\theta, \phi}$ aim to deceive $D_\psi$. The loss function for $\theta$ is:
>
> $L_{\theta} = - E_{x_1 \sim \rho_1}[D_\psi \left( H_\theta(t, x(t), \gamma_\phi(t, z)) \right) ], \quad x(t) = x_0 + \gamma_{1, \phi}(t,z) \odot x_1, \quad z \sim \rho_T,$
>
> where $L_\theta$ indicates that the gradient is only calculated with respect to $\theta$. Since $\theta$ only needs to handle elements in $\{\gamma_\phi(t, x_T) \mid t \in [0, T], x_T \sim \rho_T\}$ and not other elements in $\Gamma_h$, $\theta$ is trained exclusively on $\gamma_\phi$, helping reduce the load on $H_\theta$. The loss functions for $\phi$ and $\psi$ are:
>
> $L_\phi = - E_{x_1 \sim \rho_1}[D_\psi \left( G(\tau, x_1,  v_{\theta, \phi}) \right) ],$
>
> $L_\psi = E_{x_0 \sim \rho_0}[\max(0, 1 - D_\psi(x_0))] + E_{x_1 \sim \rho_1}[\max(0, 1 + D_\psi(G(\tau, x_1,  v_{\theta, \phi})))].$
>
> Here, $L_\phi$ and $L_\psi$ indicate that gradients are calculated with respect to $\phi$ and $\psi$, respectively, as shown in Figure 3. With these loss functions, $\phi$ is optimized to find better trajectories, while $\theta$ adapts to $\gamma_\phi$, which is sparser than $\Gamma_h$.
> "
>
> By making these adjustments, we hope to improve the clarity and accessibility of our explanations.
>
> [$\textbf{Weakness 3.}$] Limitations and Theoretical Insights: The paper briefly mentions the lack of a theoretical foundation for the effectiveness of MTO. This could be addressed by offering some initial theoretical insights. Discussing the computational complexity of training models with multidimensional coefficients and potential limitations in scalability would also be valuable.
>
> $\textbf{[3-1. Insights of MTO and adversarial approach to MTO]}$
>
> Yes, we agree with your comment and want to highlight the insights of MTO by comparing it with simulation-based modeling. If we use the 'adaptive multidimensional coefficient' specifically explained in [1-1. Comparing more rigorously with other methods with additional definition], we can adaptively utilize different coefficients for all inference trajectories. This concept has been explored in previous works like NeuralODE [8]. However, the adaptability and flexibility of trajectories inherent in these approaches were lost when adopting simulation-based objectives in diffusion models.
>
> Our adversarial approach to MTO revives these properties—previously achievable only through end-to-end simulation dynamics—while addressing training efficiency issues, as discussed in [1-2. Highlighting the practical benefits of the adversarial approach to MTO with additional experiments and observations]. To emphasize these insights, we have added the following illustration to the introduction section:

---

> ### Author Response · Authors · 2024-11-17
>
> "
> “Given a differential equation solver with fixed configurations, which multidimensional trajectory yields optimal performance in terms of final transportation quality for a given starting point of the differential equation?” This question highlights a trade-off inherent in diffusion models, where simulation-free objectives—while beneficial in reducing training costs—limit adaptability in trajectory optimization concerning output quality. This flexibility is retained in simulation-dynamics . To enable trajectory optimization while maintaining simulation-free objectives for training cost efficiency, prior approaches have relied on pre-defined trajectory properties, such as straightness [8, 9], to minimize numerical error. However, such pre-defined properties for trajectories deviate from true transportation optimality, as they fail to consider the sole measure of optimality in our perspective: the final quality of transportation, which can be calculated using simulation-dynamics.
>
> ...
>
> To effectively leverage the adaptability and flexibility of trajectories offered by simulation-based objectives while mitigating inefficiencies in training, we propose using simulation-based objectives only for $\phi$ after pre-training $\theta$ with a simulation-free objective. This makes trajectory optimization feasible in terms of efficiency and scalability. Our experiments demonstrate that trajectories optimized through this approach significantly improve the performance of flow and diffusion models.
> "
>
> By including these additional paragraphs, we hope to better convey the insights of MTO and the adversarial approach to MTO to readers with improved clarity. Thank you for your insightful points.
>
> $\textbf{[3-2. Scalability issues of adversarial approach to MTO]}$
>
> We addressed this topic in [1-2. Highlighting the practical benefits of the adversarial approach to MTO with additional experiments and observations].
>
> [$\textbf{Weakness 4.}$] Insufficiently related works: Another weakness of the paper is the insufficient presentation of related works. The few and one-step diffusion distillation is a pretty mature research direction. However, the author fails to mention a lot of well-established approaches such as DMD [1], UFOGen [2], Diffusion-GAN [3], Diff-Instruct [4], iCT [5], GET-onestep [6], TRACT [7], etc. Besides, the authors compare MTO on FFHQ and AFHQ datasets with a limited comparison with broader domains of other few-step generative models. I am curious why the authors do not conduct experiments on the well-established ImageNet-64 generation benchmark.
>
> $\textbf{[4. Insufficient related work]}$
>
> We addressed this topic in [1-1. Comparing more rigorously with other methods with additional definition].
>
> [$\textbf{Question 1.}$] What is the intuition of the objective in Equation (14).
>
> $\textbf{[1. Insight of $L_\theta$]}$
>
> Since $\theta$ only needs to handle elements in $\{\gamma_\phi(t, x_T), t \in [0, T], x_T \sim \rho_T\}$ and not other elements in $\Gamma_h$, $\theta$ is trained exclusively on $\gamma_\phi$, helping reduce the load on $H_\theta$. We added this sentence in the adversarial training section for better clarity. Thank you for your precise point.
>
> [$\textbf{Question 2.}$] I think it would be good if the authors could show the performances on the ImageNet64 benchmark.
>
> $\textbf{[2. Additional experiment on ImageNet-64 dataset]}$
>
> We apologize for the limited experiments. We completely agree with your point. We tried our best to validate our methodology's scalability and effectiveness but couldn't afford to train the EDM-based $H_\theta$ from scratch, which requires 32 A100 GPUs and several weeks of training for a fair comparison. We sincerely aim to experiment with our methods on large datasets like ImageNet and video datasets in future work.
>
> [$\textbf{Question 3.}$] Is MTO a direct generation of diffusion-induced GAN? What is the technical novelty behind MTO?
>
> $\textbf{[3. Technical novelty of MTO]}$
>
> As we mentioned in [1-1. Comparing more rigorously with other methods with additional definition], MTO differs from distillation methods or existing trajectory optimization methods in four aspects ((i), (ii), (iii), (iv)) discussed in [1-1. Comparing more rigorously with other methods with additional definition]. We have highlighted our technical novelty in [1-1. Comparing more rigorously with other methods with additional definition] and [1-2. Highlighting the practical benefits of the adversarial approach to MTO with additional experiments and observations] on the practical aspects.
>
> Overall, we sincerely thank you for your valuable insights and precise comments. Based on your feedback, we have revised our paper and uploaded a new version to reflect your comments as thoroughly as possible. If you have any further questions, we are happy to address them without hesitation.
>
> Thank you again.

---

> ### Author Response · Authors · 2024-11-18
>
> [1] Raghav Singhal, Mark Goldstein, and Rajesh Ranganath. Where to diffuse, how to diffuse, and how to get back: Automated learning for multivariate diffusions, 2023.
>
> [2] Grigory Bartosh, Dmitry Vetrov, and Christian A. Naesseth. Neural flow diffusion models: Learnable forward process for improved diffusion modelling, 2024.
>
> [3] Bowen Zheng and Tianming Yang. Diffusion models are innate one-step generators, 2024.
>
> [4] Dongjun Kim, Chieh-Hsin Lai, Wei-Hsiang Liao, Naoki Murata, Yuhta Takida, Toshimitsu Uesaka, Yutong He, Yuki Mitsufuji, and Stefano Ermon. Consistency trajectory models: Learning probability flow ODE trajectory of diffusion, 2024.
>
> [5] Ian Goodfellow, Jean Pouget-Abadie, Mehdi Mirza, Bing Xu, David Warde-Farley, Sher-
> jil Ozair, Aaron Courville, and Yoshua Bengio. Generative adversarial nets, 2014.
>
> [6] Axel Sauer, Katja Schwarz, and Andreas Geiger. Stylegan-xl: Scaling stylegan to large diverse datasets, 2022.
>
> [7] Jae Hyun Lim and Jong Chul Ye. Geometric gan, 2017.
>
> [8] Ricky T. Q. Chen, Yulia Rubanova, Jesse Bettencourt, and David K Duvenaud. Neural ordinary differential equations, 2018.
>
> [9] Alexander Tong, Kilian FATRAS, Nikolay Malkin, Guillaume Huguet, Yanlei Zhang, Jarrid Rector Brooks, Guy Wolf, and Yoshua Bengio. Improving and generalizing flow-based generative models with minibatch optimal transport, 2024.
>
> [10] Xingchao Liu, Chengyue Gong, and qiang liu. Flow straight and fast: Learning to generate and transfer data with rectified flow, 2023.
>
> [11] Michael Samuel Albergo, Nicholas Matthew Boffi, Michael Lindsey, and Eric Vanden-Eijnden. Multimarginal generative modeling with stochastic interpolants, 2024
>
> [12] One-step diffusion with distribution matching distillation
>
> [13] UFOGen: You Forward Once Large Scale Text-to-Image Generation via Diffusion GANs
>
> [14] Diffusion-GAN: Training GANs with Diffusion
>
> [15] Diff-Instruct: A Universal Approach for Transferring Knowledge From Pre-trained Diffusion Models
>
> [16] Improved Techniques for Training Consistency Models
>
> [17] One-step diffusion distillation via deep equilibrium models
>
> [18] Tract: Denoising diffusion models with transitive closure time-distillation
>
> [19] Yang Song, Prafulla Dhariwal, Mark Chen, and Ilya Sutskever. Consistency models, 2023.

---

> ### Comment · Reviewer_ZX4x · 2024-11-24
> **Thanks authors for the responses**
>
> ## Thanks authors for the responses.
>
> First of all, I would like to clarify that I did not hold **positive attitude** in the review, claiming neither novel nor impactful. **So please do not announce anything on behalf of me**!
>
> After reading the authors' rebuttal, **some of my concerns**, such as the math part of the definition of the multidimensional coefficient and the insufficiently related works, **have been resolved**. However, three important concerns still remain.
>
> (1) Literaturely, the $H_\theta$ in equation (11) can be viewed as a direct mapping that takes a noisy image as an input and outputs predicted clean data. **Therefore, minimizing the objective (13) and (14, discriminator loss) is exactly the same as existing works, such as UFOGen**, that use a time-dependence discriminator to train a multi-level generator. The only difference between MTO and UFOGen is that MTO trains an additional parametric coefficient function. From this point of view, I do not see much novelty behind such a simple and direct generalization.
>
> (2) As I have mentioned to the reviewer, the GDD-I is a special case of adversarial training (i.e. a one-step UFOGen and MTO without parametric coefficient). However, the 5-step EDM-MTO is significantly worse than the 1-step GDD-I on CIFAR10 conditional generation, FFHQ-64 and AFHQv2-64. **It is very weird that a **general multi-step model** can not outperform its simple yet specific counterpart**.
>
> (3) Besides, the EDM paper has released their ImageNet-64 checkpoint, so **I do not understand why authors claim that they have to **pre-train** an ImageNet-64 EDM model again in the rebuttal period** before they can evaluate MTO on ImageNet-64 generation benchmark.
>
> I still think the novelty behind the MTO is not that strong when compared with the existing GAN-based approach. The key point that does make sense is that the multi-step MTO shows significantly worse performances than a simple and special case (the GDD-I) on multiple benchmarks.
>
> **I will consider raising the score if the authors can provide stronger empirical evidence that demonstrates the benefits of using multi-step and coefficient optimization.**

---

> > ### Author Response · Authors · 2024-11-25
> > **Thanks reviewer for the responses**
> >
> > First of all, thank you for your concise and core feedback. If any repetition of our work caused unnecessary use of your time, we sincerely apologize for this. We would like to promptly address the misconception raised in part (3) regarding the additional experiment for ImageNet-64:
> >
> > "(3) Besides, the EDM paper has released their ImageNet-64 checkpoint, so I do not understand why authors claim that they have to pre-train an ImageNet-64 EDM model again in the rebuttal period before they can evaluate MTO on ImageNet-64 generation benchmark."
> >
> > $\textbf{Our EDM-based diffusion model structure, $H_\theta(t, x(t), \gamma)$, differs from the EDM model $H_\theta(t, x(t))$}$ in terms of "coefficient labeling $\gamma$," as explained in "Section 4.3 Coefficient Labeling for Flow and Diffusion" in our paper. Therefore, $\textbf{we cannot use the existing EDM checkpoint and must train the model from scratch}$. This is the reason why we carefully design the hypothesis space $\Gamma_h$ for the pre-training stage.
> >
> > For better readability, we have relocated this paragraph from Section 4.4 to the pre-training section (Section 4.3) of our paper. Thank you for your insightful feedback.
> >
> > We will work diligently to provide additional empirical evidence before the rebuttal period concludes.

---

> > > ### Comment · Reviewer_ZX4x · 2024-11-25
> > > **Thanks author's response**
> > >
> > > Thank you for the response.
> > >
> > > It seems that the MTO requires pre-train diffusion models from scratch (instead of reusing existing diffusion models). This property may prevent MTO from being widely applied when pre-trained diffusion models exist. **I think it would strengthen the submission if authors could somehow figure out a way to re-use existing diffusion models for MTO**.

---

> ### Author Response · Authors · 2024-11-26
> **Thanks Reviewer ZX4x for the response**
>
> Thank you for the response and for suggesting a direction to improve MTO.
>
> We entirely agree with your point that it is cumbersome to train flow or diffusion models from scratch to perform MTO. This should be addressed as one of the main limitations of our research for future researchers. We have revised the limitation section as follows to highlight this issue and propose a potential solution:
>
> "First, since we use coefficient labeling (Section 4.3) for the diffusion model, the model structure differs from existing pre-trained flow and diffusion models. As a result, training models from scratch is required, which can be cumbersome. This issue could be mitigated in future works by replacing a few layers from a well pre-trained model and using it as initialization for pre-training. ..."
>
> Thank you again for your quick and concise response.

---

> ### Author Response · Authors · 2024-11-27
> **Thanks for Reviewer ZX4x for the response (Extra experimentation results are uploaded in supplementary materials)**
>
> $\textbf{Q. I will consider raising the score if the authors can provide stronger empirical evidence that demonstrates the benefits of using multi-step ...}$
>
> We would like to emphasize the contribution of the additional $\phi$ parameterization and its optimization, which is supported by extra experimental results in the supplementary materials.
>
> We want to highlight that all the experiments in our supplementary materials for MTO are $\textbf{obtained by training only $\phi$, while keeping $\theta$ frozen}$. As shown in the supplementary materials, particularly in the SI results on CIFAR-10 and ImageNet-32, we achieved comparably good FID values using only a few steps. These additional experimental results demonstrate that SI and MTO are a good combination, potentially sparking new interest in SI methodology. While these FID values are not close to those of current SOTA methodologies, this is due to the limitations of FM and SI and optimizing only $\phi$ rather than limitations of MTO itself, given that we demonstrated SOTA performance (CIFAR-10 conditional) using EDM in the main paper.
>
> Overall, the results in the supplementary materials show that we achieved on-par performance by optimizing only $\phi$ with an adversarial objective across various methodologies. This underscores the novelty and effectiveness of our approach in leveraging $\phi$ optimization for trajectory adaptability and improved generative performance.
>
> Thank you for your valuable responses.

---

> ### Author Response · Authors · 2024-11-28
> **Appendix E (Additional Experiment): Organizing and clarifying experiments in supplementary materials**
>
> We have added Appendix E: Additional Experiment in our main paper to organize and clarify the experiments presented in the supplementary materials. This section includes detailed descriptions of network architectures, training configurations, and concise discussions for each training result.
>
> We want to highlight three key points:
>
> 1. Pre-training with the multidimensional coefficient not only preserves but also enhances the performance of flow and diffusion models.
> 2. All MTO experiments in Appendix E were conducted by training only $\phi$, while keeping $\theta$ frozen.
> 3. Even with a small 4-layer discriminator network and Vanilla GAN Loss, we achieved comparable FID performance on SI.
>
> Thank you for your attention to these updates.

---

> > ### Author Response · Authors · 2024-12-03
> > **Summary of Revisions and Final Check**
> >
> > We have summarized our revisions and key points in a top comment for all reviewers. If there are any additional clarifications or points you’d like us to address before the discussion closes, please let us know. Thank you again for your thoughtful feedback.

---

> > > ### Comment · Reviewer_ZX4x · 2024-12-03
> > > **Thanks authors for the response**
> > >
> > > Thank the authors for the response and efforts during the rebuttal period.
> > >
> > > However the approach still has essential issues, such as the requirement of pre-training instead of reusing existing diffusion models, I would like to raise my score to 6 to encourage authors for deeper explorations.

---

> > > > ### Author Response · Authors · 2024-12-03
> > > > **Thanks reviewer ZX4x for the response**
> > > >
> > > > Thank you for your thoughtful feedback and for raising the score. We sincerely appreciate your recognition of our efforts during the rebuttal period. We acknowledge the issue you highlighted regarding the need for pre-training diffusion models from scratch instead of leveraging existing pre-trained models. Addressing this challenge will be one of the key focuses in our future work. Your encouragement motivates us to further refine and expand the applicability of our approach. Thank you once again.

---

### Author Response · Authors · 2024-12-03
**Summarization of the Rebuttal for Everyone**

We sincerely thank all the reviewers for their valuable comments on our paper. We have diligently addressed most of the concerns raised by the reviewers with various additional experiments. Below, we summarize the major revision points:

$\textbf{[1. Novelty of extra $\phi$ parameterization (Reviewer ZX4x and EetL)]}$

Reviewers ZX4x and EetL raised concerns that the GAN-based objective function itself is not novel, and the empirical benefits of extra $\phi$ parameterization and multi-step generation seem limited. We address these concerns as follows:

1. Our novel $\phi$ parameterization, termed the "Adaptive Multidimensional Coefficient," is essential to removing trajectory constraints for flow and diffusion modeling. By this, our work is $\textbf{the first work enabling true "Trajectory Optimization" for flow and diffusion models.}$

2. Although GAN-based objectives aren't novel, $\textbf{our approach is non-trivial:}$ using distinct objectives for $\theta$ and $\phi$, achieving remarkable efficiency and performance as explained below.

3. By optimizing trajectories, we achieved $\textbf{SOTA performance in CIFAR-10 conditional generation}$ (FID = 1.37) with low NFE (=5). Additional experiments showed that this performance $\textbf{could not be achieved by CTM modeling (FID = 1.98, 2.04 for NFE = 5, 6)}$, a popular SOTA method. This highlights the benefits of small multi-step (NFE = 5) generation, first explored in our work, compared to existing one-step or two-step approaches.

4. In Appendix E, additional experiments validate two empirical benefits for Stochastic Interpolants, Flow Matching, and DDPM on CIFAR-10 and ImageNet-32. First, $\textbf{training with multidimensional coefficients not only maintains but enhances model performance.}$ Second, $\textbf{we achieved comparable performance (FID = 4.14 on CIFAR-10) across frameworks by optimizing only $\phi$}$ with low NFE (=10) (Also for our 2-dimensional experiments). These results strongly support the empirical benefits of the "Adaptive Multidimensional Coefficient" (extra $\phi$ parameterization).

$\textbf{[2. Related works (Reviewer ZX4x and dLdR)]}$

We revised the related works section to compare our methodology specifically against all the works suggested by the reviewers. This revision highlights the distinctions and contributions of our method, addressing concerns regarding its novelty.

$\textbf{[3. Training efficiency and scalability (Reviewer EetL and dLdR)]}$

We observed that $\textbf{our methodology exhibits remarkable training efficiency,}$ requiring 5–10 times fewer kimg compared to GDD, a method known for its efficiency. These results, included in our revisions, further emphasize the scalability of our approach.

$\textbf{[4. Paper clarity, particularly for adaptive multidimensional coefficient and adversarial training (Reviewer ZX4x, xR7Z, and dLdR)]}$

We revised all sections identified by the reviewers to improve clarity. Specifically:
- Added Definition 2 for the coefficient, Equation 7 for hypothesis space, and Equations 10 and 12 for the adversarial training section.
- Revised the abstract, introduction, methodology, and conclusion for better comprehension.
These updates aim to concisely explain our methodology without ambiguity.

$\textbf{[5. Limitations and Future Works (Reviewer ZX4x, EetL, and xR7Z)]}$

We agree with the reviewers' suggestions about limitations, which are:
1. Training a flow or diffusion model from scratch is cumbersome.
2. Fixed sampling configurations for inference limit flexibility.
3. Results on FFHQ and AFHQv2 are less promising.

We respond as follows:
1. Additional results in Appendix E demonstrate that $\textbf{training flow and diffusion models with a multidimensional coefficient from scratch can improve performance}$. $\textbf{This property can be adapted to all the flow and diffusion modeling}$. Future work could mitigate the cumbersome issue by partially leveraging pre-trained models.
2. $\textbf{The "trajectory optimization" problem is inherently defined for "fixed sampling configurations" (e.g., NFE)}$. This represents a trade-off between performance and flexible inference scheduling. $\textbf{Our work is the first to highlight this trade-off}$, which can be further explored in future work by conditioning $\gamma_\phi$ on diverse sampling configurations.
3. We want to highlight that this is not a limitation of MTO, as $\textbf{this work represents the first attempt to design the hypothesis space}$ for "Multidimensional Coefficient" and "Adaptive Multidimensional Coefficient." $\textbf{There are numerous potential design choices for this,}$ and future work can explore and address these possibilities.

We will address these limitations by refining training strategies, leveraging pre-trained models, and exploring flexible trajectory designs in future work.

We thank the reviewers for their valuable insights and believe these revisions address the concerns raised. Thank you for your attention.

---

### Meta-Review · Area_Chair_f2XT · 2024-12-22

**Metareview:**

The paper proposes a generalization of diffusion models by introducing multi dimensional coefficients for the schedule; this is followed by a second stage of adversarial training that optimizes for the optimal coefficients given a NFE budget. Experiments seem to support the effectiveness of the method. The paper receives borderline ratings. All reviewers agree that this is an interesting idea, but raised concerns regarding the novelty and complexity of the adversarial training, as well as clarify in presentation. The authors provided thorough responses to the questions, which helped to address some of the concerns. However, not all reviewers are convinced that all their concerns are addressed. After going through the paper and discussions, the AC believes that this is indeed a promising work but is not quite ready in its current shape. I believe that an improved version of this work would make it a solid contribution to the field and encourage the authors to submit it to the next venue.

**Additional Comments On Reviewer Discussion:**

Reviewers raised concerns regarding the novelty and complexity of the adversarial training, as well as clarify in presentation. The authors provided thorough responses to the questions, which helped to address some of the concerns. However, not all reviewers are convinced that all their concerns are addressed

---

### Decision · Program_Chairs · 2025-01-22

Reject